# Transmetalation in Cancer Pharmacology

**DOI:** 10.3390/ijms262211008

**Published:** 2025-11-14

**Authors:** Mahendiran Dharmasivam, Busra Kaya

**Affiliations:** Institute for Biomedicine and Glycomics, Griffith University, Southport, QLD 4215, Australia

**Keywords:** transmetalation, thiosemicarbazone, copper/iron pharmacology, lysosomal targeting, reactive oxygen species, ferroptosis

## Abstract

Transmetalation, the exchange of metal ions between coordination complexes and biomolecules, has emerged as a powerful design lever in cancer metallopharmacology. Using thiosemicarbazones (TSCs) as a unifying case study, we show how redox-inert carrier states such as zinc(II) or gallium(III) can convert in situ into redox-active copper(II) or iron(III/II) complexes within acidic, metal-rich lysosomes. This conditional activation localizes reactive oxygen species (ROS) generation and iron deprivation to tumor cells. We critically compare redox-active and redox-inert states, delineating how steric and electronic tuning, backbone rigidity, and sulfur-to-selenium substitution govern exchange hierarchies and kinetics. We further map downstream consequences for metal trafficking, lysosomal membrane permeabilization, apoptosis, and ferroptosis. Beyond TSCs, iron(III)-targeted transmetalation from titanium(IV)-chelator “chemical transferrin mimetics” illustrates a generalizable Trojan horse paradigm. We conclude with translational lessons, including mitigation of hemoprotein oxidation via steric shielding, stealth zinc(II) prodrugs, and dual-chelator architectures and outline biomarker, formulation, and imaging strategies that de-risk clinical development. Collectively, these insights establish transmetalation as a central therapeutic principle. We also highlight open challenges such as quantifying in-cell exchange kinetics, predicting speciation under non-equilibrium conditions, and rationally combining these agents with existing therapies.

## 1. Introduction

Metal-based therapeutics have long been cornerstones of cancer treatment, with platinum drugs such as cisplatin demonstrating the clinical potential of inorganic chemistry in oncology [1,2,3,4,5,6,7]. Beyond platinum, a rich landscape of ligands and complexes has been explored to exploit dysregulated metal homeostasis in tumors [1,2,3,4,5]. Among these, thiosemicarbazones (TSCs) and their derivatives have gained prominence as agents that simultaneously chelate essential metals and generate cytotoxic redox activity [3,5,8,9,10]. TSCs bind biologically relevant metals including iron (Fe), copper (Cu), and zinc (Zn), and this underlies their therapeutic versatility [3,5,8,9,10].

Notably, di-2-pyridylketone thiosemicarbazones (DpT; Figure 1A), for example, di-2-pyridylketone-4,4-dimethyl-3-thiosemicarbazone (Dp44mT; Figure 1B) and di-2-pyridylketone-4-cyclohexyl-4-methyl-3-thiosemicarbazone (DpC; Figure 1C), are highly redox-active in their Fe(III), and especially Cu(II) complex forms, generating reactive oxygen species (ROS) that induce apoptosis in tumor cells [8,9,11,12]. This dual-mechanism ‘metal sequestration plus redox attack’ termed the ‘double-punch’ strategy, is a hallmark of TSC pharmacology [8,10,13]. Crucially, recent studies have revealed that transmetalation, the exchange of metals between complex and surrounding biomolecules, plays a pivotal role in modulating both the efficacy and selectivity of these metal-based drugs [5,8,9,10,13,14,15,16,17,18].

Transmetalation can occur when a metal–ligand complex encounters a competing metal ion with higher binding affinity for the ligand [8,9]. In the context of cancer therapy, this means an administered metal complex might exchange its metal for another metal ion inside the intracellular environment, effectively activating or deactivating certain pathways [8,9]. For example, many thiosemicarbazone complexes are designed to be stable while circulating yet undergo metal exchange upon entering the metal-rich intracellular milieu of tumor cells [8,9]. This strategy exploits the dependence of cancer cells on metals such as iron and copper for proliferation, while minimizing off-target interactions in the bloodstream [5]. Given the complexity of the tumor microenvironment including aberrant metal metabolism, acidic organelles, and oxidative stress understanding transmetalation at a mechanistic level is critical [8,9].

This review integrates chemical and biological perspectives on transmetalation in cancer pharmacology, with particular emphasis on metal-based ligand systems, especially thiosemicarbazones, and their binding to titanium(IV) (Ti(IV)), Fe(III), cobalt(III) (Co(III)), nickel(II) (Ni(II)), Cu(II), Zn(II), and palladium(II) (Pd(II)) [5,8,9,10,13,14,18,19]. We compare redox-active and redox-inert complexes in terms of anticancer efficacy and toxicity. Next, we highlight advances in ligand design, including steric and electronic tuning, backbone modifications, and chalcogen (sulfur(S)/selenium (Se)) isosteric substitution. We then examine biological implications such as lysosomal targeting, ROS-mediated cell death, ferroptosis, and interference with oncogenic signaling. Finally, we consider translational aspects, surveying how insights from recent studies, including those from our group, are guiding the preclinical and clinical development of next-generation metal-based anticancer agents.

## 2. Metal-Based Ligand Systems in Cancer Therapy: Thiosemicarbazones and Beyond

Thiosemicarbazones as dual-action agents

Thiosemicarbazones represent a versatile class of chelating agents that have been investigated for decades due to their potent and selective antitumor activity [5,8,9,10,13,14,18,20,21]. These compounds coordinate transition metals through nitrogen–nitrogen–sulfur (*N*, *N*, *S*; Figure 1) donor sets, forming stable complexes that disrupt metal-dependent processes in cancer cells [5,8,9,10,13].

A prototypical example is Triapine^®^ (3-aminopyridine-2-carboxaldehyde-thiosemicarbazone; Figure 1D), a drug that entered clinical trials as an inhibitor of ribonucleotide reductase (RNR) [22,23,24]. Triapine^®^ chelates intracellular iron and forming a redox-active Fe(II) complex that quenches the essential tyrosyl radical in RNR’s active site [23,25,26,27]. This deprives cancer cells of deoxyribonucleotides needed for DNA replication, inducing S-phase arrest and apoptosis [28,29,30,31].

Beyond Triapine^®^ (Figure 1D), extensive structure-activity relationship studies on DpT (Figure 1A) analogues have yielded first- and second-generation agents, such as Dp44mT (Figure 1B) and DpC (Figure 1C), with improved potency and in vivo tumor selectivity [11,12,32,33,34,35,36]. These agents not only sequester iron to mimic an ‘acute iron starvation’ in cancer cells, but their metal complexes also catalyze ROS production, a combined effect underlying their broad anticancer efficacy [11,12]. However, this potency came with drawbacks. For example, the Fe(III) complex of Dp44mT oxidizes oxy-hemoglobin to met-hemoglobin and oxy-myoglobin to met-myoglobin in vitro and in vivo, contributing to cardiac toxicity [37].

The later analogue DpC was designed with a bulky cyclohexyl substituent to mitigate this effect, and it does not induce cardiotoxicity in vivo [37]. This reflects DpS’s superior safety profile. Building on this, third-generation thiosemicarbazones such as (*E*)-3-phenyl-1-(2-pyridinyl)-2-propen-1-one-4,4-dimethyl-3-thiosemicarbazone (PPP44mT; Figure 1E) were developed to further overcome these limitations [8]. More recent generations, (*E*)-3-phenyl-1-(2-pyridinyl)-2-propen-1-one-4-phenyl-3-selenosemicarbazones (PPP4pSe; Figure 1F) have been discovered that completely abrogate oxy-myoglobin oxidation, thereby preventing the muscle pain observed in patients, while retaining anticancer efficacy [9,10,13].

### 2.1. Beyond TSCs Other Metal–Ligand Systems

While thiosemicarbazones are a focal point, other metal-binding ligand systems have also been explored. For instance, hydroxamates and catecholates, such as those found in the iron chelator (Figure 2A–C) desferrioxamine (Figure 2A) or enterobactin (Figure 2B) analogues, sequester iron to inhibit tumor growth, whereas gallium complexes such as gallium maltolate (Figure 3A) and tris(8-quinolinolato)gallium(III) (Figure 3B) hijack iron-metabolism pathways by substituting for Fe(III) in biological systems [38,39,40,41,42,43,44,45,46,47,48,49,50,51,52,53,54]. Notably, enterobactin, a bacterial catecholate siderophore, has been repurposed in oncology: enterobactin and synthetic analogues deplete intracellular Fe(III) and suppress proliferation by iron starvation and redox modulation, illustrating how microbial iron-acquisition chemistry can be leveraged against tumors [45,46,47]. Related naturally occurring siderophores, including desferrioxamine and pyoverdine (Figure 2C), similarly perturb tumor iron homeostasis and provide bioinspired templates for anticancer chelators [48,49,50,51,52,53,54].

Early non-platinum complexes like titanium(IV) cyclopentadienyls (e.g., titanocene dichloride) introduced novel mechanisms of action, though issues such as aqueous instability hampered their development [55,56,57,58,59]. Collectively, these examples highlight a key lesson: the success of metal-based therapies often depends on controlling metal speciation and reactivity in vivo, a challenge that transmetalation-based design aims to address.

### 2.2. Transmetalation as a Design Principle

Our focus remains on TSCs and related systems where transmetalation is leveraged for therapeutic gain [5,8,9,10,13,19]. In these designs, the administered metal complex acts as a prodrug that undergoes metal exchange in the target environment to yield the active form [5,10]. This concept was demonstrated by studies showing that a presumably inert Zn(II)–thiosemicarbazone complex can ‘switch on’ its cytotoxic activity by exchanging Zn for Cu inside cells [8,9,14]. For example, [Zn(DpC)_2_] localizes to lysosomes and, under acidic lysosomal conditions, transmetallates with Cu(II) to form a redox-active Cu-DpC complex (Figure 4). The newly formed Cu complex triggers lysosomal membrane permeabilization (LMP) and cell death [14].

Representative transmetalation reactions:[Zn(DpC)_2_] + Cu^2+^ ⇋ [Cu(DpC)_2_] + Zn^2+^(1)[Zn(DpC)_2_] + Cu^2+^ ⇋ [Cu(DpC)]^+^ + Zn^2+^ + DpC^−^(2)

These Equations (1) and (2) illustrate the neutral and mono-ligand pathways, respectively, by which Zn(II)–thiosemicarbazone complexes convert into redox-active Cu(II) species under physiological conditions [9].

Similarly, Ti(IV) complexes bearing high-affinity Fe(III) chelators (‘‘Fe(III) trap’’ complexes) have been designed to remain intact during circulation but transmetalate with intracellular Fe(III), releasing Ti(IV) and forming Fe(III)–ligand complexes in situ [18,27]. This targeted dissociation confines the toxic effects largely to cancer cells, which possess higher labile iron pools and a reducing intracellular environment absent in normal tissue [18,27]. Overall, metal-based ligand systems in cancer therapy exemplify the synergy between coordination chemistry and biology. By tuning ligand structures, metal preferences, and redox properties, one can maximize tumor-specific damage while sparing normal tissue.

### 2.3. Transmetalation Mechanisms Among Fe(III), Cu(II), Zn(II), and Ti(IV)

Transmetalation refers to the process by which one metal in a complex is replaced by another metal ion [5,8,9,10,13,14,18,19,27]. In a biological context, this can occur when a drug complex encounters a metal ion with higher binding for the ligand, or when environmental factors such as pH or redox potential favor ligand exchange. Among Fe(III), Cu(II), Zn(II), and Ti(IV), all relevant in cancer pharmacology, distinct transmetalation scenarios arise, each with therapeutic implications:

#### 2.3.1. Transmetalation of Zn(II) to Cu(II)

Zinc is redox-inert and forms kinetically labile complexes with many thiosemicarbazones, while the same ligands generally bind Cu(II) more strongly [5,8,9,13,14,19,60,61,62,63,64,65,66,67,68,69,70,71]. In cells, especially in acidic lysosomes, Zn–thiosemicarbazone complexes can exchange Zn(II) for Cu(II), creating redox-active Cu–thiosemicarbazone species that catalyze ROS production and trigger LMP and cell death (Figure 4) [14]. This Zn(II) to Cu(II) metal switch has been directly visualized and functionally linked to cytotoxicity for Zn(II) complexes of di-2-pyridylketone thiosemicarbazones, which accumulate in lysosomes and transmetalate with available Cu(II) [14].

Recent metal-dependence studies with the DpT analogue di-2-pyridylketone-4-methyl-4-ethyl-3-thiosemicarbazone (Dp4e4mT) clearly demonstrated this principle (Figure 5A–G) [5]. Preformed Zn(II) and Ga(III) complexes of Dp4e4mT rapidly and completely exchanged with Cu(II) under biologically relevant conditions (Figure 5F,G), yielding low-nanomolar antiproliferative potency identical to the preformed Cu(II) complex [5]. In contrast, complexes that resist this exchange, such as Co(III) and Pd(II) congeners (Figure 5B,C), remain far less active, underscoring that transmetalation is a key activation step [5].

The same design logic extends beyond the DpT scaffold. For example, *N*-acridine thiosemicarbazones (NATs) were developed to target lysosomes. Their isolated Zn(II) complexes promptly transmetalate with Cu(II), maintaining redox activity while also suppressing unwanted oxy-myoglobin oxidation compared to earlier series [10]. Independently, Zn(II) complexes of SIRTi1/2-inspired thiosemicarbazones were shown to transmetalate to Cu(II) inside cells and induce ROS-mediated paraptosis, again highlighting Zn(II) as a prodrug handle delivering the ligand to the compartment where Cu(II) activates it [67].

Mechanistically, both thermodynamics and medium effects drive the exchange. Cu(II)–thiosemicarbazone complexes are thermodynamically favored, and lysosomal acidity accelerates Cu-catalyzed thiol oxidation that fuels redox cycling of Cu– thiosemicarbazones [8,11]. Cellular ligands like glutathione (GSH) and metallothionein further modulate speciation, promoting reduction in Cu(II) and ligand exchange that channel Zn–thiosemicarbazones toward Cu–thiosemicarbazones [13,19,72]. Outside cells, protein partners also matter: human serum albumin can first capture Zn(II) and then slowly deliver Cu(II), enabling Zn(II) to Cu(II) transmetalation of bis(thiosemicarbazone) complexes in extracellular media. This helps explain extracellular and pericellular activation routes [68].

Collectively, these studies support a practical prodrug concept [5,8,9,10,13]. In essence, one can deliver the ligand as a stable, redox-inert Zn(II) complex that localizes to lysosomes, then rely on endogenous Cu(II) pools to generate the active Cu–thiosemicarbazone species in situ [8,9]. This couples compartment-specific activation with potent, ROS-mediated killing and allows medicinal chemistry modifications to the ligand or metal center to balance activation, off-target reactivity, and safety [5,8,9,10,14].

#### 2.3.2. Transmetalation of Fe(III)/Fe(II) to Cu(II)

Iron and copper often compete for TSC ligands [5,8,9,10,12,13,15,20,32,36]. Many thiosemicarbazones can chelate either Fe(III) (or Fe(II)) or Cu(II), and the predominant complex in cells may depend on metal availability and redox state [5,8,13,15]. Fe(III)–ligand complexes can undergo partial transmetallation if Cu(I/II) is present (Figure 5A), especially under reducing conditions that convert Fe(III) to Fe(II) [5]. However, evidence suggests Cu(II) generally outcompetes Fe(III) for strong thiosemicarbazone ligands only if kinetic lability permits. For example, one study found that Ni(II) and Co(III) complexes of Dp4e4mT did not efficiently transmetallate with Fe(III) in solution (Figure 5B,D), whereas Zn(II) and Ga(III) complexes did exchange with Fe(III) to form [Fe(Dp4e4mT)_2_]^+^ albeit more slowly than their exchange with Cu (Figure 5F,G) [5].

In cellular environments, the labile iron pool (largely Fe(II)) might be the primary target for chelation by free ligands or incoming complexes [19,73]. For example, if a Cu-thiosemicarbazone complex enters a cell, it may reduce Fe(III) to Fe(II) and subsequently swap the Cu for Fe forming a Fe–ligand complex while releasing Cu [5]. The biological outcome of Fe vs. Cu complex formation can differ. Fe(II)-thiosemicarbazone complexes participate in Fenton chemistry and inhibit iron-dependent enzymes like RNR, whereas Cu(I/II) complexes may produce broader oxidative damage to biomolecules [74,75,76,77,78,79,80,81]. Thus, Fe-Cu transmetallation in cells represents a dynamic equilibrium that can modulate the drug’s activity spectrum [82].

#### 2.3.3. Transmetalation of Ti(IV) to Fe(III)

Titanium(IV) is not a native bio-metal but has been employed as a carrier in “Trojan horse” strategies [27,83]. For example, Ti(IV) complexes bearing high-affinity Fe(III) chelators such as deferasirox (Figure 6), an FDA-approved iron chelator have been designed [18,27,83]. These Ti-chelator complexes remain intact in extracellular conditions (Ti(IV) is relatively stable when bound to strong O-donor ligands), but once they permeate cancer cells, intracellular Fe(III) displaces Ti(IV) from the chelator [27,83]. This transmetalation releases Ti(IV) (which may then engage in independent cytotoxic actions) and simultaneously forms Fe(III)–ligand complexes inside the cell [27,83]. Notably, this transmetalation is driven by the acidic, reducing intracellular milieu: Ti(IV) tends to hydrolyze and is more easily displaced under these conditions [27,83].

Consistent with hard–soft acid–base (HSAB) principles, deferasirox (Figure 6) is an O-donor tridentate ligand that binds Fe(III) more strongly than Ti(IV), making Fe(III) substitution thermodynamically favored once inside the cell [83,84]. This preference underpins the observed Fe(III) to Ti(IV) transmetalation behavior in biological systems [83,85].

The Fe(III) that binds the ligand is drawn from the cell’s labile iron pool, meaning the drug effectively hijacks cellular iron [18]. An illustrative outcome was observed with a Ti(IV)-deferasirox complex: upon Fe uptake it yielded [Fe(deferasirox)_2_]^3−^, which inhibited RNR activity, while the liberated Ti(IV) species could directly bound nucleotide phosphates and cleave them [27,83]. This dual mechanism exploits transmetalation to exert a multi-faceted attack on cancer metabolism [18,27,83].

#### 2.3.4. Transmetalation of Zn(II) to Fe(III)

Although Zn(II) and Fe(III) differ in charge and preferred coordination geometry, some thiosemicarbazones can bind both (often as tetrahedral Zn(II) complexes and octahedral Fe(III) complexes) [5,8]. In biological systems, a Zn–thiosemicarbazone complex might also capture Fe if Zn dissociates [5,8]. The kinetics are typically slower than for Cu exchange; for instance, the conversion of [Zn(Dp4e4mT)_2_] to [Fe(Dp4e4mT)_2_]^+^ required prolonged incubation with FeCl_3_ in solution (Equation (3); Figure 7) [5]. Nonetheless, gradual Zn to Fe transmetallation can occur, especially in acidic compartments where Zn–ligand bonds weaken. This pathway may be relevant in iron-overloaded tumor regions or within endosomes/lysosomes that contain labile iron from ferritin turnover.[Zn(Dp4e4mT)_2_] + Fe^3+^ ⇋ [Fe(Dp4e4mT)_2_]^+^ + Zn^2+^(3)

Each of these transmetalation scenarios underscores a guiding principle: the affinity hierarchy and lability of metal–ligand bonds determine the fate of the complex in vivo. Hard, trivalent metals like Ga(III) or Co(III) form very stable complexes that resist exchange, useful for delivering ligands intact, but potentially limiting activation [5,15]. In contrast, softer divalent metals (Cu(II), Fe(II)) or hydrolyzable metals (Ti(IV)) can be displaced more readily [5,8,9,10,13,19,27]. Effective drug design often involves choosing a metal–ligand combination that holds together until reaching the tumor site, then hand off the ligand to a target metal or vice versa upon a specific trigger. In the following sections, we explore how such transmetalation events impact downstream redox chemistry, the trafficking of metals in cells, and the selective toxicity towards cancer cells.

## 3. Impact of Transmetalation on Redox Activity, Metal Trafficking, Tumor Selectivity and ROS Generation

The therapeutic benefit of many metal complexes derives from redox chemistry, the ability to cycle between oxidation states and produce ROS that damage biomolecules [86,87]. Transmetalation can sharply modulate this redox behavior by altering which metal is bound to the ligand [8,9]. Thiosemicarbazone complexes illustrate this vividly. A complex of a thiosemicarbazone with Zn(II) or Ga(III) is largely redox-inert; Zn(II) and Ga(III) are redox-stable and do not participate in Fenton-type reactions [5]. However, once such a complex transmetalates to Cu(II) or Fe(II/III), its reactivity changes [5]. Cu(II)-thiosemicarbazone complexes readily undergo redox cycling (Cu(II)/Cu(I)) in the presence of cellular reductants and O_2_, generating ROS such as superoxide and hydroxyl radicals [8,9,11]. To clarify redox trends, representative redox potentials are summarized in Table 1; only Fe^3+^/Fe^2+^ and Cu^2+^/Cu^+^ lie within the biological redox window, enabling Fenton-like ROS formation, while Zn^2+^/Zn and Ga^3+^/Ga remain redox-stable and Ti^4+^/Ti^3+^ is only weakly active.

Similarly, Fe(III)-thiosemicarbazone complexes can be intracellularly reduced to Fe(II)-thiosemicarbazone, which then reacts with peroxide to produce highly reactive radicals [19]. ROS generation is a double-edged sword: it can cause extensive oxidative damage to cancer cell components (DNA, lipids, proteins), but it can also harm normal cells if not controlled. Transmetalation effectively serves as a switch that turns on ROS generation only in the intended context. For example, the Zn(II) complex of a cyclohexyl-substituted thiosemicarbazone, (1-(pyridin-2-yl)-3-(p-tolyl)prop-2-en-1-one-4-cyclohexyl-4-methyl-3-thiosemicarbazone (PPTP4c4mT), a styrene-appended analogue was relatively benign until it exchanged Zn for Cu in lysosomes, upon which the newly formed Cu(II) complex triggered robust ROS-mediated cytotoxicity [9]. By keeping the complex in a redox-inactive form during circulation and only allowing redox activation inside target cells, transmetalation-based design can enhance the therapeutic index of such compounds [5,8,9,10,13].

### 3.1. Metal Trafficking and Homeostasis

Displacement of a metal from a complex does not eliminate the metal; rather, the released ion can engage in downstream biological processes with consequential effects [19,88]. In Zn to Cu transmetalation, the liberated Zn(II) may be sequestered by metallothioneins or effluxed via zinc transporters [89,90]. Meanwhile, the chelator ligand now carrying Cu(II) can redistribute this copper within the cell [91]. Thiosemicarbazones have been shown to redirect Cu into lysosomes an organelle often considered a “metal sink” or storage depot [8,9,11].

In fact, cancer cells treated with certain thiosemicarbazones accumulate redox-active Cu in lysosomes, leading to lysosomal membrane permeabilization (LMP) and a form of Cu-dependent lysosomal cell death [11,14]. This phenomenon is related to ferroptosis and underscores how transmetalation influences metal localization. Likewise, the Ti(IV) released in a Ti to Fe transmetalation may not remain innocuous: Ti(IV) has a strong affinity for oxygen-donor biomolecules (especially phosphates) [27,57]. It could bind to Adenosine Triphosphate (ATP) or deoxyribonucleic acid (DNA) backbone phosphate groups, potentially disrupting metabolism or triggering localized DNA damage.

Thus, transmetalation cause a non-trivial rerouting of metal ions. A therapeutic complex might deliver one metal (like Ti or Zn) to a specific compartment and concurrently hijack an essential metal (Fe or Cu) from that compartment. This perturbation in metal trafficking can amplify stress in cancer cells, which already operate under higher basal metal requirements and oxidative load.

### 3.2. Tumor Selectivity

One might fear that transmetalation could occur indiscriminately in healthy tissues, but there are several reasons why these processes show tumor-selective effects. Key factors include:

#### 3.2.1. Elevated Intracellular Metals in Tumors

Several cancer models exhibit elevated labile iron and increased free copper levels associated with hyperactive metabolism and upregulated metal-import pathways. For example, breast (MCF-7, MDA-MB-231) [92,93], prostate (PC-3, DU145) [94], and neuroepithelioma (SK-N-MC) [94] cells show higher TfR1 and CTR1 expression and greater labile Fe and Cu levels compared with non-malignant counterparts [95,96,97,98]. In contrast, normal cells tightly regulate and minimize free metal ions [99]. Therefore, in such contexts, a transmetalation-dependent drug encounters more endogenous “fuel” for activation, enhancing selectivity [100].

#### 3.2.2. Acidic and Reducing Tumor Compartments

Many tumors exhibit aberrant acidity in endosomes and lysosomes, as well as greater baseline oxidative stress [101]. These conditions promote complex dissociation and metal exchange. For example, lysosomal sequestration of a Zn–thiosemicarbazone complex followed by exchange with Cu is far more likely in cancer cells because of their heightened endocytic activity and tendency to funnel copper into lysosomes (via upregulated copper transporters like ATP7A/B) [14,102]. The outcome is selective formation of the active Cu complex in cancer lysosomes, sparing normal cells [14].

#### 3.2.3. Differential Signaling Responses

Downstream effects of metal chelation favor cancer cell death [103]. Iron chelators like thiosemicarbazones trigger tumor-specific signaling changes, for instance, upregulating the metastasis suppressor protein N-myc downregulated gene-1 (NDRG1) and downregulating pro-proliferative factors that normal cells do not strongly exhibit [8,9,13]. By exchanging metals in situ, the drug exploits the tumor’s metal addiction, often described as “metabolic hijacking”. Notably, a recent dual-chelator conjugate, deferasirox *N*-ethyleneamine triapine (DefNEtTrp; Figure 8), showed much higher cytotoxicity toward cancer cells than healthy cells, presumably because it capitalizes on the abundant Fe(II/III) in cancer cells to activate its two chelation moieties [18].

In summary, transmetalation can be harnessed to concentrate cytotoxic, metal-based reactions within tumor cells by leveraging differences in metal handling between malignant and normal tissues [100]. Careful design and selection of the carrier metal and ligand ensure that the “trigger” conditions for metal exchange (excess Cu or Fe, low pH, reducing environment) are predominantly encountered in the tumor microenvironment [8,9,14].

## 4. Redox-Active Versus Redox-Inert Complexes: Efficacy and Off-Target Toxicity

A recurring theme in metallodrug design is balancing between redox activity (for potent tumor kill) and stability or inertness (for safety) [104]. Complexes can be broadly classified as redox-active (capable of cycling between oxidation states and generating radicals) or redox-inert (maintaining a single oxidation state and not directly producing ROS) [5,8,9,10,13,14]. Each category has distinct implications for anticancer efficacy and toxicity:

### 4.1. Redox-Active Complexes

Examples include Cu(II) or Fe(III) complexes of thiosemicarbazones [5,8,9,10,13,14]. These tend to have high anticancer efficacy because they induce oxidative stress, damage DNA, and trigger cell death pathways [11,12]. For instance, Cu(II) complexes of thiosemicarbazones like Dp44mT or DpC efficiently redox-cycle and can cause rapid tumor cell death (Figure 9), one study observed cytotoxic effects within only a few hours of exposure to a Cu–thiosemicarbazone complex [11,12,36]. Moreover, redox-active complexes can directly target redox-sensitive enzymes (Triapine^®^-Fe(II) complex inactivates RNR by ROS-mediated radical quenching) [18,31,105].

However, this potency comes at a cost, as uncontrolled redox activity often leads to off-target toxicity. A well-known example is the oxidation of hemoglobin and myoglobin by Fe(III)–thiosemicarbazone complexes [5,8,9,10,13,37]. As mentioned earlier, the Fe(III) complex of first-generation thiosemicarbazone, Dp44mT oxidizes oxy-hemoglobin to met-hemoglobin and oxy-myoglobin to met-myoglobin in vivo [37], contributing to cardiotoxicity [106]. Such off-target oxidation in blood and muscle can impair oxygen delivery and damage tissues. Likewise, redox-active copper complexes might cause systemic oxidative stress or injury to the liver and kidneys, which clear metals [107,108,109,110]. Thus, while redox-active complexes are often extremely effective at killing cancer cells, they require built-in control mechanisms to avoid collateral damage, either through selective activation (as with transmetalation) or through dosing strategies that exploit cancer’s heightened ROS vulnerability [8].

In addressing clinical translation, formulation and delivery strongly influence toxicity outcomes [111]. Redox-active Cu(II) and Fe(III) complexes are generally administered intravenously in animal and human studies to avoid gastrointestinal degradation and ensure bioavailability [112]. Their intrinsic redox activity can lead to hemoglobin oxidation and oxidative organ stress; however, careful dose optimization and the use of redox-inert or “stealth” analogues such as Zn(II)- and Ga(III)-thiosemicarbazones mitigate these effects [8]. Moreover, formulation approaches including encapsulation in liposomes or PEGylated nanocarriers have been shown to reduce systemic toxicity while preserving tumor selectivity [113].

### 4.2. Redox-Inert Complexes

Examples include Zn(II) or Ga(III) complexes of the same ligands [5,8,9,10,13,114,115]. These complexes do not themselves engage in redox cycling, meaning they produce little to no ROS directly (Figure 9) [5,8,9,10,13,115]. In terms of off-target effects, this inertness is beneficial, for example, a Zn(II)-thiosemicarbazone complex will not oxidize hemoglobin, and indeed new thiosemicarbazone analogues have been designed such that their Fe(III) or Zn(II) complexes do not readily oxidize oxy-myoglobin, thus mitigating potential cardiotoxicity [5,8,9,10,13]. The trade-off is that redox-inert complexes can be less immediately potent, essentially acting as a “dormant” form of the drug.

The concept of a “stealth” complex has been introduced to describe this behavior. A Zn(II) complex of a thiosemicarbazone, [Zn(PPP44mT)_2_], where PPP44mT is a styryl-substituted ligand showed a pronounced delay in antiproliferative effect, requiring ~48 h to achieve tumor cell kill, whereas the free ligand or its Cu complex acted within 3 h [8]. This delayed action is attributed to the need for the Zn complex to either dissociate or transmetalate to a Cu or Fe complex inside cells before the drug becomes fully active [8].

In essence, the Zn complex “sneaks” into the cell without causing immediate damage, then unfolds its activity over time, hence the “stealth” moniker [8]. The advantage is a likely reduction in acute toxicity. Indeed, the same study found that the stealth [Zn(PPP44mT)_2_] complex caused significantly less immediate oxidative stress to normal biomolecules, yet ultimately it effective in suppressing oncogenic signaling, for example, down-regulating the cell-cycle driver cyclin D1 more potently than inherently active Cu complexes [8]. The disadvantage is that if conversion to an active form is inefficient or too slow, the overall therapeutic effect may be diminished or variable.

In practice, the distinction between “redox-active” and “redox-inert” is not absolute, many complexes can exist in equilibrium with small amounts of free ligand or partially substituted species that contribute some ROS [107,116,117]. Nonetheless, these categories are useful for guides. Ideally, an anticancer complex would travel as a redox-inert species to avoid off-target damage and then convert into a redox-active form specifically inside the tumor [118].

This goal is evident in multiple modern strategies. For example, Ga(III) complexes of thiosemicarbazones are being studied as stable prodrugs that release their active ligands upon encountering transferrin-bound iron in tumors [5,50,119,120,121]. Similarly, Co(III) complexes, which are kinetically inert low-spin d^6^ systems, have been proposed as carriers that release cytotoxic ligands within the reducing environment of hypoxic tumor tissue, where Co(III) can be enzymatically reduced to the more labile Co(II) state under low oxygen conditions [122,123]. Ultimately, comparing redox-active and inert complexes underscores a key design principle: achieve sufficient reactivity to kill cancer cells, but contain that reactivity until the drug is in the right place [124,125,126]. Transmetalation, as discussed, is one powerful means to this end, effectively turning a redox-inert prodrug into a redox-active drug on site [5].

## 5. Advances in Ligand Design for Transmetalation Control and Efficacy

Researchers have developed sophisticated ligand modifications to fine-tune when and where transmetalation occurs, as well as to modulate the downstream reactivity of metal complexes [5,8,9,10,13,16,67,83,127,128,129]. Key design strategies include steric hindrance, electronic effects, backbone rigidity, isosteric atom substitution and multi-functional chelation:

### 5.1. Steric Tuning (Hindrance and Bulk)

Introducing bulky substituents on a ligand can dramatically influence the behavior of its metal complexes [9,10]. One benefit of steric bulk is the shielding of the metal center from unwanted interactions [9,10]. For example, adding a bulky cyclohexyl group to Dp44mT to develop DpC was a breakthrough, the steric encumbrance is thought to prevent the planar [Fe(DpC)_2_]^+^ complex from approaching the heme center of hemoglobin or myoglobin, thereby reducing its ability to oxidize these off-target proteins [8,9,10].

Similarly, adding a styryl moiety to a thiosemicarbazone, PPP44mT, not only enhanced cytotoxic potency but also sterically hindered the complex’s approach to heme groups [8,9]. The new styryl-substituted complexes showed no measurable oxidation of oxy-myoglobin, even as they retained potent anticancer activity.

Steric bulk can also slow down ligand exchange kinetics. While this might seem counterintuitive for promoting transmetalation [8,9]. With careful design, steric bulk can block unwanted interactions, such as protein binding or off-target dimerization yet still allow exchange with target metals at a confined coordination site [9,10]. In essence, steric tuning imposes size selectivity: it preserves transmetalation with small metal ions yet suppresses macromolecular interactions like protein docking or heme association.

### 5.2. Electronic Tuning (Donor Strength and Redox Potential)

Substituents on the ligand can donate or withdraw electron density, altering the ligand’s metal-binding affinity and the redox properties of its complex [8]. A clear example is the incorporation of an electron-rich acridine moiety into a thiosemicarbazone framework creating NATs [10,130]. The acridine group extends conjugation and increases electron donation to the metal, shifting the Fe(III)/Fe(II) redox couple to more positive potentials in NAT-Fe complexes compared to analogous non-acridine complexes [10].

In practical terms, NAT-Fe(III) complexes are harder to reduce (less prone to spontaneously form Fe(II) and generate ROS), a beneficial trait for avoiding random oxidative damage [10]. Yet, once inside cancer cells, these complexes still undergo reduction and participate in cytotoxic redox cycling, just with a bit more control. Electronic effects can also dictate metal preference. Ligands with softer donor atoms or greater polarizability favor softer metals (Cu(II), Hg(II), etc.), whereas hard donors favor hard metals (Fe(III), Al(III), etc.) [131,132,133]. By tuning electronic properties, chemists can bias a ligand to exchange metals in a certain direction.

For instance, adding electron-donating groups can stabilize a metal–ligand bond, reducing the likelihood of dissociation until a strong competitor is present [134,135]. Conversely, electron-withdrawing substituents can weaken the bond, making the complex more labile and ready to release or swap its metal [136,137]. In the realm of thiosemicarbazones, subtle electronic tuning has been used to adjust the pKa of coordinating atoms and the overall complex stability, thereby influencing at the pH or redox conditions under which transmetalation occurs.

### 5.3. Backbone Rigidity and Conformation

The flexibility or rigidity of a ligand backbone can affect how a complex interacts with biomolecules and whether it can accommodate multiple coordination geometries during metal exchange. A rigid, planar ligand (e.g., one containing extended aromatic systems) often forms geometrically constrained complexes [10,138,139]. This can be advantageous: a rigid ligand may hold a metal very tightly in a specific geometry, but if that metal is removed, the ligand might quickly rearrange around a new metal in a similarly stable fashion, facilitating transmetalation rather than complete dissociation.

In contrast, a very flexible ligand might adapt to suboptimal binding, forming various adducts or oligomers, which could either impede or unpredictably alter transmetalation pathways [140]. In the case of thiosemicarbazones, adding ring systems (pyridine, phenyl, acridine) increases rigidity and planarity [10,12,36,130]. The *N*-acridine thiosemicarbazones mentioned earlier not only tuned electronics but also created a flat, aromatic platform that likely encourages π–π stacking with DNA and perhaps easier intercalation, potentially helping drag the complex to DNA-rich regions and then swap metals there for a targeted effect [10,130].

Additionally, rigidity can influence lipophilicity. More planar aromatic systems tend to increase hydrophobic character, aiding passive diffusion through cell membranes up to a point. Indeed, the NAT Zn(II) complexes were noted to be highly lipophilic and readily taken up by cells, effectively acting as “chaperones” for their own delivery [10]. Once inside, their ability to dissociate or transmetalate yielded the active Cu complexes, aligning with the stealth delivery concept. Thus, backbone design is a delicate balancing act: it must allow the necessary chemistry (metal binding and releasing) while imparting favorable pharmacokinetic properties (cell permeability, subcellular localization).

### 5.4. Isosteric Substitution (Sulfur Versus Selenium and Beyond)

A powerful approach in medicinal chemistry is to swap an atom in the molecular framework for a closely related isosteric to probe changes in activity [9,13]. In thiosemicarbazones, replacing sulfur with selenium in the thiosemicarbazone backbone forming selenosemicarbazones has shown striking effects [9,13]. Selenium, directly below sulfur in the periodic table, has similar bonding patterns but is more polarizable and slightly larger. A recent study compared a thiosemicarbazone (PPTP4c4mT) with its selenium analogue, (1-(pyridin-2-yl)-3-(p-tolyl)prop-2-en-1-one-4-cyclohexyl-4-methyl-3-selenosemicarbazone) PPTP4c4mSe and found that the selenium substitution enhanced anticancer efficacy and tumor selectivity, largely due to augmented transmetalation [9]. The Zn(II)–selenosemicarbazone complex exhibited greater propensity to dissociate and exchange with Cu(II) in lysosomes than its Zn(II)–thiosemicarbazone counterpart, leading to more ROS generation and cytotoxicity (Figure 10) [9].

This can be rationalized by selenium’s softer donor character [9,13]. It binds Zn(II) less tightly (facilitating Zn release) while still forming robust complex with Cu(II) (Cu^2+^ being borderline-soft and bonding strongly with selenium) [141]. Consistent with this, the selenosemicarbazone’s superior cytotoxicity was attributed to its enhanced lysosomal Cu transmetalation and resultant increase in ROS [9].

Additionally, the selenium-containing ligand showed subtle differences in biological interactions, selenium can engage in different hydrogen-bonding or polar interactions compared to sulfur, potentially affecting cellular distribution and target engagement [142,143,144]. Beyond sulfur to selenium, other isosteric or isoelectronic substitutions (e.g., oxygen vs. sulfur in certain chelating units, or –CH= vs. –N= in heterocycles) are being explored to fine-tune metal affinity sequences and transmetalation kinetics [145,146]. Each such modification can tilt the balance of how and when a metal complex might transform in the body.

### 5.5. Multi-Functional Ligand Design (Dual Chelator)

In tandem with these strategies, multi-functional ligand design has gained attraction. One example is the dual chelator DefNEtTrp (Figure 8), which covalently links two different metal-binding pharmacophores: an aroylhydrazone (deferasirox-like) unit for Fe(III), and a thiosemicarbazone (Triapine-like) unit for Fe(II)/Cu(II)) [18]. This design ensures that no matter the oxidation state of intracellular iron, at least one site on the ligand will capture it.

The dual chelator can coordinate one Fe ion with both arms, or two Fe ions simultaneously at distinct sites [18]. Mechanistically, such a ligand could undergo sequential transmetalation, first exchanging an initial placeholder metal for one iron ion, then scavenging another iron with its second site [18]. The result is cooperative, multidentate binding that strongly anchors iron inside cancer cells, shifting the equilibrium towards drug-bound iron and away from cellular metalloproteins. Initial biological testing of DefNEtTrp showed potent activity and the ability to induce both apoptosis and ferroptosis, validating the idea that bridging different chelator motifs can amplify therapeutic mechanisms [18]. Contemporary ligand design for metallopharmaceuticals is highly tunable: small changes in structure yield significant differences in transmetalation behavior, redox profiles, and biological interactions.

## 6. Biological Implications: Lysosomal Targeting, ROS, and Cell Death Pathways

The downstream biological effects of transmetalation and metal complex dissociation are multifaceted [5,8,9,10,13,14]. Here, we detail how these chemical events drive cellular outcomes, including lysosomal trapping, oxidative stress, apoptosis, ferroptosis, and disruption of oncogenic signaling:

### 6.1. Lysosomal Trapping and Activation

Many metal complexes accumulate in lysosomes, acidic organelles known for recycling and storage [8,9,14]. Thiosemicarbazones and similar cationic complexes often become protonated and sequestered in lysosomes due to pH partitioning (weak bases concentrate in acidic compartments) and active transport by metal pumps. This accumulation is not merely passive; lysosomes appear to be critical “reaction chambers” where transmetalation and redox chemistry unfold [8,9,14]. For example, Zn(II)–thiosemicarbazones complexes tend to transmetalate with Cu(II) specifically in lysosomes, as shown by co-localization studies and the requirement of lysosomal acidity for maximal cytotoxicity (Figure 4 and Figure 10) [14]. Within the lysosome, Cu(II) can be released from metallothioneins or imported via transporters like ATP7B, providing the substrate for exchange.

The newly formed Cu(thiosemicarbazone)_2_ complex can then redox cycle to produce ROS inside the lysosome [8,9,14]. Because lysosomal membranes are rich in lipids and lysosomes lack robust antioxidant defenses, they are vulnerable to peroxidative damage. The result is often LMP, spilling acidic and proteolytic contents into the cytosol [14]. LMP is a known trigger of the intrinsic (mitochondrial) apoptosis pathway and can also lead to necrotic cell death if extensive. Furthermore, iron-dependent ROS generation in lysosomes can initiate ferroptosis, an iron-catalyzed form of cell death characterized by catastrophic lipid peroxidation.

Indeed, some thiosemicarbazones (like Triapine) have been shown to induce ferroptosis in cancer cells, likely through excessive redox-active iron in lysosomes or cytosol. Intriguingly, the dual induction of apoptosis and ferroptosis by certain chelators (e.g., DefNEtTrp) suggests that lysosomal iron perturbation (ferroptotic signaling) can coexist with classical caspase-dependent pathways [18]. This multifaceted cell death induction is highly desirable in cancer therapy, as it may overcome resistance mechanisms that allow tumor cellsto escape any single mode of death.

### 6.2. ROS Generation and Oxidative Damage

As discussed, transmetalation often leads to the formation of redox-active complexes that produce ROS [5,8,9,10,13,14]. The biological implications of ROS go beyond simply “killing cells.” At sub-lethal levels, ROS can act as signaling molecules, for instance, activating stress pathways (p38 MAPK, NRF2 antioxidant responses, DNA damage responses via ATM/ATR) which can lead to growth arrest or adaptive survival [147,148,149,150].

However, at the high levels produced by potent metal complexes, ROS inflict irreparable damage: DNA strand breaks, lipid membrane oxidation, protein unfolding, and depletion of reducing equivalents (NADPH, GSH) [108,151,152]. Key cellular targets include mitochondria (loss of membrane potential and release of cytochrome c, an apoptosis trigger) and DNA (causing replication stress and activation of p53) [153,154]. The ROS burst from Cu-thiosemicarbazone complexes has been directly linked to mitochondrial dysfunction and caspase activation in cancer cells [155].

Notably, ROS can also inactivate specific iron-dependent enzymes (apart from RNR), such as aconitase in the tricarboxylic acid (TCA) cycle, compounding the metabolic crisis in the cancer cell [103,156,157,158]. The concept of an “oxidative shock” to cancer cells is attractive, but the challenge is mitigating it in normal cells, hence the emphasis on controlled delivery and conditional transmetalation [159,160,161]. Importantly, some of the new ligand designs (e.g., those with bulky or electron-donating groups) aim to produce ROS in a localized and transient manner, enough to kill the cell from within, but not so diffusely that neighboring healthy cells or blood components are harmed [162].

### 6.3. Apoptosis and Cell Cycle Effects

Many metal chelators and complexes induce programmed cell death (apoptosis) in cancer cells via multiple mechanisms [103,163,164,165,166]. Iron deprivation triggers DNA replication arrest (through RNR inhibition and activation of the intra-S-phase checkpoint), while ROS can trigger DNA damage checkpoints or directly activate pro-apoptotic factors [29,167]. Upstream, iron depletion caused by chelators leads to stabilizes of hypoxia inducible factor (HIF-1α) and activates p53, both of which push cells toward apoptosis when damage is sufficient [168,169,170]. Downstream, as noted, release of cytochrome c from mitochondria (often due to oxidative damage and LMP) leads to caspase cascade activation [171,172,173]. Several studies have documented hallmark apoptotic indicators (caspase-3 cleavage, Annexin-V staining, DNA fragmentation) after treating tumor cells with metal-binding ligands or their complexes [174,175,176,177,178].

Furthermore, thiosemicarbazones like Dp44mT and DpC cause marked cell cycle arrest at G_1_/S or S phase due to the loss of iron-dependent cell cycle proteins (e.g., cyclin D1) and stalled replication forks [179,180,181]. This synchronized arrest can make cells more susceptible to apoptosis if the damage is not repaired. The combination of cell cycle arrest (cytostatic) and apoptosis (cytotoxic) means these agents can both stop tumor growth and reduce tumor mass. The aim is to tip the balance fully toward irreversible cell death in cancer cells, while causing at most a temporary cell-cycle delay in normal cells from which they can recover once the drug is cleared.

### 6.4. Ferroptosis

A distinct iron-dependent form of non-apoptotic cell death, ferroptosis, is garnering attention in the context of metal-based drugs [182,183,184,185,186]. Ferroptosis is driven by overwhelming lipid peroxidation and failure of GSH-dependent peroxide repair via glutathione peroxidase 4 (GPX4; Figure 10) [187,188,189,190,191]. While classical ferroptosis inducers (like erastin; Figure 11) block cystine uptake (depleting GSH), metal chelators can induce ferroptosis by a different route: flooding the cell with redox-active iron [192,193,194,195]. This seems paradoxical, since chelators remove iron; however, certain chelators (like Triapine or Dp44mT) can redox-cycle iron and increase the pool of Fe(II) [156,181,196,197].

For example, Triapine’s Fe(III) complex readily reduces to Fe(II) and, in doing so, may bypass some of the cell’s iron-sequestering mechanisms [23,25]. The net effect is similar to iron overload in specific compartments, promoting Fenton chemistry on lipid membranes. Some thiosemicarbazones also downregulate ferritin (the iron storage protein) or upregulate transferrin receptor, inadvertently causing iron mismanagement that feeds ferroptosis.

The dual chelator DefNEtTrp mentioned earlier was explicitly shown to cause features of ferroptosis (e.g., cell death rescued by lipophilic antioxidants, and dependence on iron) in addition to caspase-dependent apoptosis [18]. The ability to trigger ferroptosis is significant because ferroptosis can kill even those cancer cells that have evaded apoptosis (e.g., via p53 mutation or Bcl-2 overexpression) [198,199,200]. Conversely, ferroptosis in normal tissues (e.g., neurons) is implicated in degenerative diseases [201,202,203]. This underscore that tumor-targeted activation of the transmetalation/ROS process is crucial to avoid unintended ferroptosis in healthy cells.

### 6.5. Inhibition of Oncogenic Signaling

Beyond direct killing, metal chelation therapy induces a cascade of cellular signaling changes. Iron depletion by chelators is known to upregulate NDRG1, a metastasis suppressor. NDRG1 has pleiotropic effects: it negatively regulates several oncogenic signaling pathways including phosphoinositide 3-kinase/protein kinase B (PI3K/AKT) and Wnt/β-catenin and reduces the activity of receptor tyrosine kinases like epidermal growth factor receptor (EGFR), human epidermal growth factor receptor 2 (HER2), mesenchymal–epithelial transition factor (MET), and Insulin-like Growth Factor 1 Receptor (IGF1R) [204,205,206,207]. By elevating NDRG1 (via iron sequestration), thiosemicarbazones effectively put the brakes on proliferation and angiogenesis signaling [207,208].

Concurrently, cyclin-dependent kinase inhibitors like p21/CIP1 can be upregulated (via p53 or as a response to DNA damage), reinforcing cell cycle arrest [209,210]. The “double punch” mechanism of thiosemicarbazones also includes direct redox attacks on signaling proteins [13,35]. For example, ROS can inactivate phosphatases that restrain kinase cascades, sometimes leading to an initial transient spike in mitogen-activated protein kinase (MAPK) or c-jun N-terminal kinase (JNK) pathway activity. Such perturbations can promote cell death or, if sublethal, trigger adaptive stress responses.

Interestingly, one of the newer thiosemicarbazone complexes, [Zn(PPP44mT)_2_], was found to suppress the oncogene cyclin D1 (a key driver of G_1_/S transition and often overexpressed in cancer) more effectively than even the clinical candidate DpC [8]. This indicates that refined metal complexes are not only killing cells outright but also reprogramming signaling networks in ways unfavorable to tumor growth. Such multi-targeted disruption of cancer cell biology is beneficial for preventing resistance: even if a cell manages to counteract one mode (e.g., oxidative stress), it might still succumb to iron starvation or loss of pro-survival signaling, and vice versa.

## 7. Translational Considerations and Clinical Outlook

Translating metal-based ligand systems from bench-to-bedside involves addressing safety, stability, formulation, and demonstrating clear advantages over existing therapies [211,212,213,214]. Several thiosemicarbazone-based agents and metal complexes have already advanced to clinical trials, offering valuable lessons for future development [36,215,216]:

### 7.1. Preclinical Efficacy vs. Toxicity

In animal models, thiosemicarbazones such as Dp44mT showed potent suppression of tumor growth but revealed toxicity at high doses (e.g., cardiac fibrosis in mice, attributed to off-target oxidative damage) [106]. The improved analogue DpC was developed to mitigate this, and in mice it achieved strong antitumor effects without cardiac toxicity [36]. DpC progressed to a Phase I clinical trial in 2016 based on its favorable efficacy and safety profile. The case of DpC underscores the translational importance of ligand modifications (like cyclohexyl substitution) to reduce toxicity while maintaining cancer selectivity [8].

Beyond oncology, several metal–thiosemicarbazone complexes also demonstrate antibacterial potential through mechanisms analogous to their anticancer activity. Cu(II) and Fe(III) complexes catalyze Fenton-type redox cycling, generating ROS that damage bacterial membranes, proteins, and DNA, while Zn(II) complexes can inhibit metalloenzymes critical for microbial survival. Incorporation of lipophilic substituents or formulation within nanocarriers enhances bacterial uptake and biofilm disruption. These properties, reported for several TSC analogues and their metal complexes, underscore the wider biomedical versatility of this chemical class and its promise for addressing antibiotic-resistant infections [184,186].

The clinical translation of metal-based complexes is closely tied to their solubility, formulation, and delivery route. Compounds such as Dp44mT have primarily been evaluated through intravenous administration [217], whereas its improved analogue DpC demonstrates high stability and oral bioavailability [36], enabling successful progression to clinical trials. Lipophilic analogues and nanoformulations now further support oral or targeted delivery options. The water solubility of thiosemicarbazone ligands varies widely; cyclodextrin inclusion, liposomal dispersion, or introduction of polar substituents (e.g., morpholine, piperazine) have been effective strategies to enhance aqueous compatibility [218,219]. Addressing intrinsic toxicity remains essential, as Cu(II) and Fe(III) complexes may exhibit oxidative reactivity, whereas Ga(III) and Zn(II) complexes act as redox-inert prodrugs with improved tolerability. Collectively, these advances balance efficacy with safety to overcome barriers to clinical translation.

Another agent, Triapine, underwent multiple clinical trials (including Phase II trials in combination with cisplatin and radiation for cervical cancer) owing to its RNR-inhibitory action [220,221]. However, patients experienced side effects like methemoglobinemia, reflecting the drug’s mechanism of oxidizing hemoglobin iron [37]. This led to explorations of modified dosing schedules and formulations for Triapine (e.g., short infusion prior to radiation, when tumor oxygenation is higher, to preferentially radiosensitize the tumor) [222]. Future trials of transmetalation-based drugs will likely incorporate biomarker monitoring, for example, checking patient blood for signs of oxidative stress or shifts in metal levels, to ensure therapeutic windows are not exceeded.

### 7.2. Pharmacokinetics and Formulation

Metal complexes often face challenges such as poor aqueous solubility, rapid plasma clearance (if they dissociate or bind serum proteins), and suboptimal tissue distribution. DpC, for instance, is hydrophobic and was delivered in preclinical studies using specialized formulations (liposomes or cyclodextrin carriers) to improve bioavailability [36]. Maintaining a complex intact until it reaches tumors can be tricky if serum proteins (albumin, transferrin) or small thiols (GSH) strip the metal prematurely.

One translational strategy is to encapsulate metal complexes in nanoparticles or liposomes, shielding them from blood components and exploiting the enhanced permeability and retention (EPR) effect in tumors [223,224,225,226]. Additionally, prodrug approaches are being considered, such as administering the ligand in a non-toxic form that will pick up a metal in vivo [227,228]. For example, one could deliver a chelator that selectively binds copper in the tumor microenvironment, essentially using endogenous copper as the activating metal (analogous to how tetrathiomolybdate “mops up” copper needed for angiogenesis) [229,230,231].

Along these lines, one could envision giving a Zn(II)-thiosemicarbazone complex as a prodrug alongside a localized copper ‘‘booster’’ (though controlling the latter is challenging) [8,10]. On the analytical front, advanced imaging techniques are being employed to track metal drugs. For instance, magnetic resonance imaging (MRI) with iron-based complexes, or positron emission tomography (PET) with radiolabeled gallium-68 or copper-64 thiosemicarbazone complexes can reveal where compounds distribute and whether they release their metal in the tumor [232,233,234,235].

### 7.3. Clinical Trial Design and Patient Stratification

As these therapies move forward, it will be important to identify patient populations who might benefit the most. Cancers with high iron metabolism (e.g., leukemias with transferrin receptor overexpression, breast cancers with elevated ferritin) could be particularly susceptible to iron-chelation strategies [95,236]. Likewise, tumors with certain mutations (e.g., RAS or p53) have altered redox states that might make them more vulnerable to ROS-inducing drugs or ferroptosis.

There is also growing interest in combining metal-binding drugs with other treatments [237,238,239]. For example, an iron chelator could be used to enhance an alkylating agent (by limiting DNA repair), or a ROS-producing complex could be paired with immune checkpoint inhibitors (since ROS-induced tumor cell damage can increase neoantigen release and immune visibility) [20,240]. Such combinations must be carefully timed and dosed to avoid excessive normal tissue harm, but the rationale is strong for tackling difficult-to-treat cancers. Importantly, patient monitoring in trials could include biomarkers of metal status (e.g., serum iron, copper, ferritin, ceruloplasmin) and oxidative stress markers to guide dosing and ensure safety.

### 7.4. Regulatory and Manufacturing Considerations

Metal-containing drugs add complexity in quality control, batches must be checked for the precise metal-to-ligand ratio, absence of free (unbound) metal or ligand, and consistency in oxidation state. For instance, Ti(IV) complexes must be manufactured under controlled conditions to prevent Ti reduction or hydrolysis before use [241]. Regulatory agencies will scrutinize issues like heavy metal accumulation (e.g., will long-term treatment cause titanium to build up in bone, or excess zinc in the liver?) and excretion pathways.

Encouragingly, many metals being considered (Fe, Cu, Zn) are essential elements with well-known homeostatic mechanisms; the key is to ensure the drug does not chronically disrupt these systems. Patients on iron chelators, for example, might require monitoring of blood iron indices to guard against anemia, though in trials of DpC, no significant changes in systemic iron were noted at effective doses [36]. Another regulatory aspect is intellectual property: numerous thiosemicarbazone derivatives are patented, which could influence which compounds advance based on commercial interests and licensing.

### 7.5. Emerging Candidates and Future Directions

Among the cutting-edge developments are multinuclear complexes (molecules carrying two different metal centers in one assembly to perform tandem functions) and targeted organometallic conjugates (in which a metal complex is attached to a tumor-seeking vector like a peptide or antibody). While not classical transmetalation, these approaches rely on related principles of controlled metal release and action. Additionally, selenium-containing thiosemicarbazones (as discussed with PPTP4c4mSe) have opened a new avenue for tuning pharmacology and may enter preclinical development given their superior in vitro profile [9].

The concept of a “stealth” complex is likely to be tested in vivo. For instance, does [Zn(PPP44mT)_2_] indeed show less acute toxicity in animal models compared to an equivalent Cu complex, and can it then activate in tumors to the same degree? Early indications are promising [8]. One study showed that certain Zn(II) complexes are unusually effective against cancer cells in vitro even without adding exogenous Cu, hinting that in the cellular milieu they switch to active forms on their own.

## 8. Conclusions

The development of metal-based anticancer complexes is being guided by continuous refinement of ligand frameworks and a deeper understanding of tumor metal biology [5,8,9,10,12,13,36,130]. The overarching aim is to deliver a decisive biochemical strike by disrupting malignant metal trafficking, generating overwhelming oxidative stress, and inducing multiple modes of cell death, while sparing healthy tissues. Transmetalation has emerged as a central design principle in this context, functioning as a molecular “switch” that conditionally activates redox chemistry in cancer’s most vulnerable compartments [5,8,9,10,13]. By exploiting differences in metal pools, lysosomal acidity, and redox tone, this approach couple’s potent chemistry with biological selectivity. Emerging generations of thiosemicarbazones, selenosemicarbazones, and multi-functional chelators demonstrate that precise structural tuning can separate efficacy from toxicity, highlighting that safety and potency can be co-engineered [9,13]. Collectively, these advances establish transmetalation as the driving mechanism of a paradigm shift in metallopharmacology, offering realistic prospects for overcoming even the most treatment-resistant cancers.

## Figures and Tables

**Figure 1 ijms-26-11008-f001:**
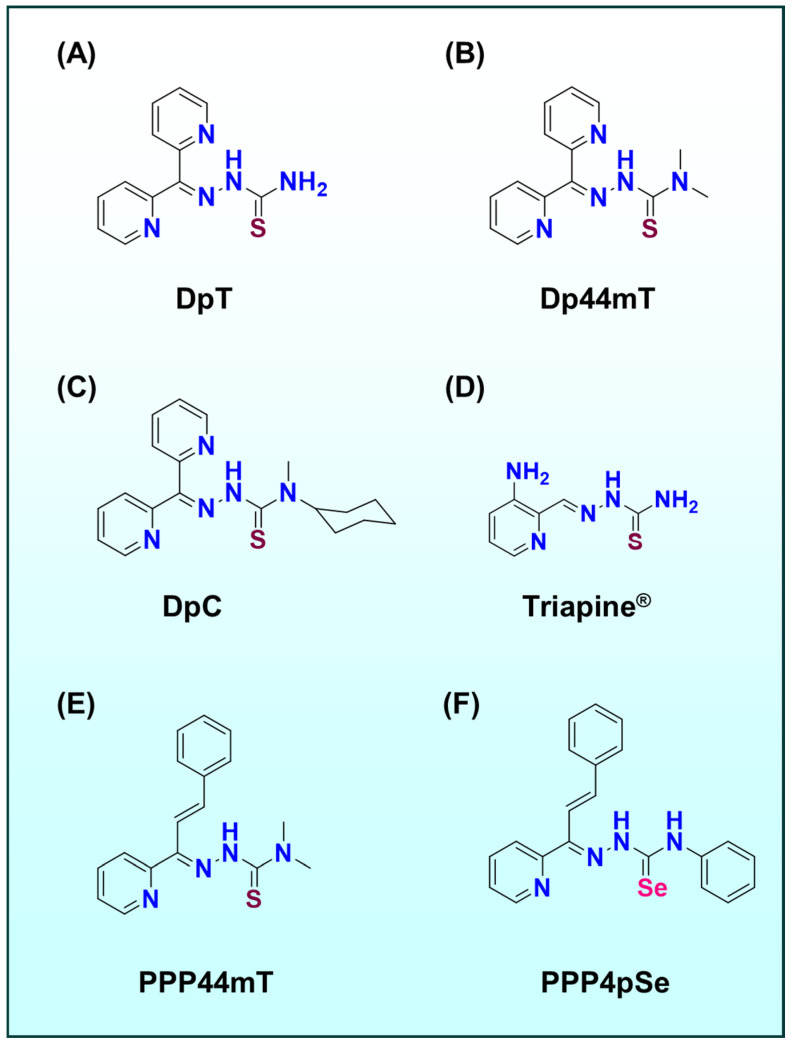
Chemical structures of (**A**) DpT, (**B**) Dp44mT, (**C**) DpC, (**D**) Triapine^®^, (**E**) PPP44mT, and (**F**) PPP4pSe.

**Figure 2 ijms-26-11008-f002:**
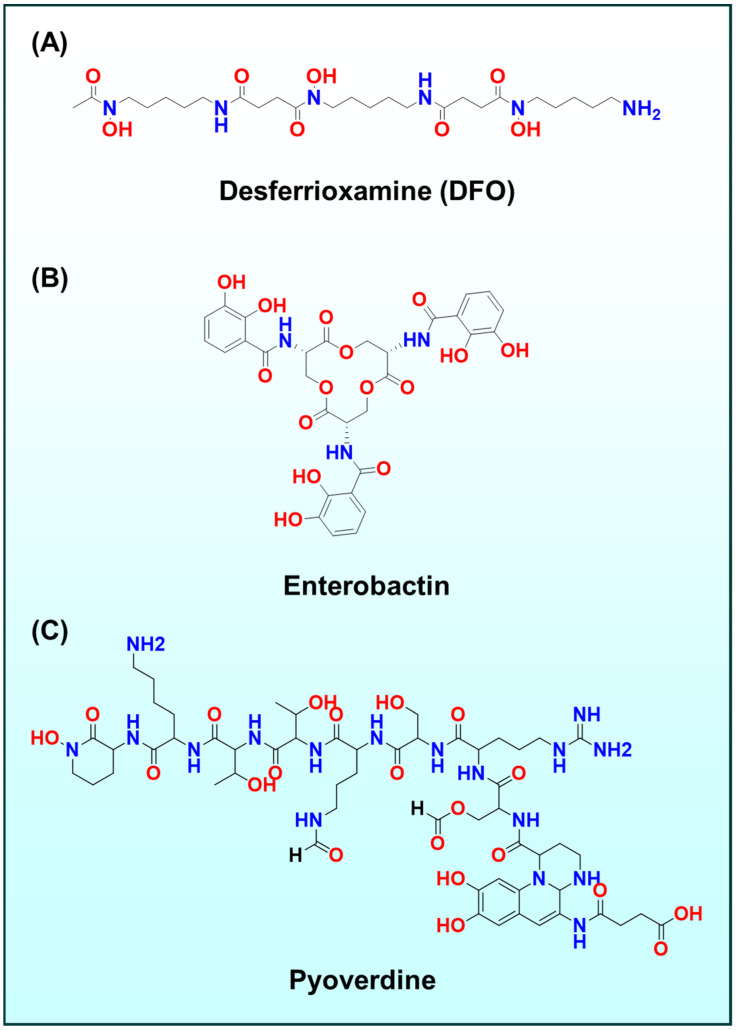
Chemical structures of (**A**) DFO; (**B**) Enterobectin; and (**C**) Pyoverdine.

**Figure 3 ijms-26-11008-f003:**
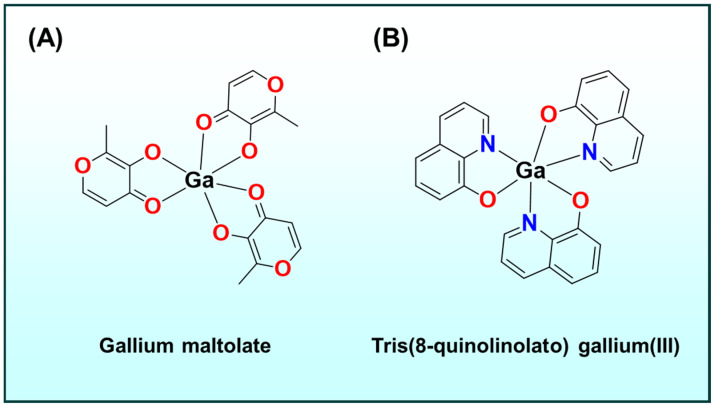
Chemical structures of (**A**) gallium maltolate; and (**B**) tris(8-quinolinolate)gallium(III).

**Figure 4 ijms-26-11008-f004:**
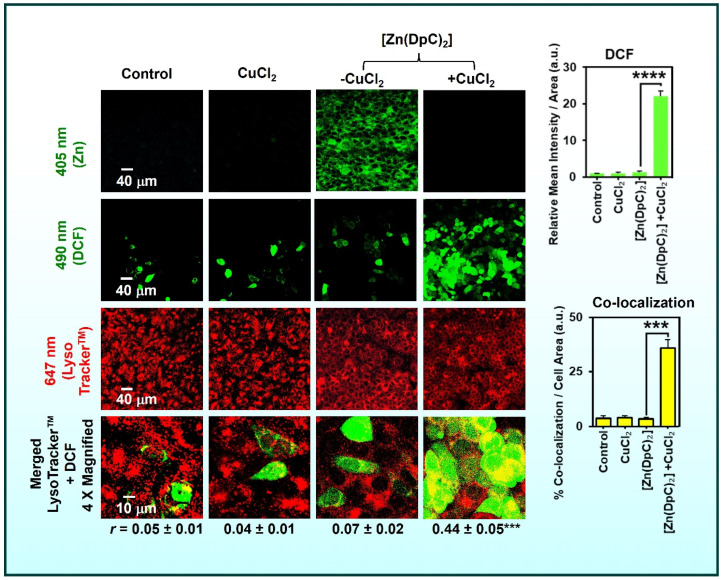
Illustration of metal exchange and lysosomal ROS generation mediated by [Zn(DpC)_2_]. In cells, [Zn(DpC)_2_] undergoes transmetalation with Cu(II), forming a redox-active Cu(II)–DpC complex that accumulates in lysosomes and promotes localized ROS generation (visualized by DCF fluorescence). The merged images show strong co-localization of ROS (green) with lysosomes (red), confirming that transmetalation triggers lysosomal oxidative stress rather than diffuse cytosolic oxidation. Scale bars: 40 µm (main panels) and 10 µm (magnified images). Results are means ± SD (3). *** *p* < 0.001; **** *p* < 0.0001 vs. the control or as indicated. Reproduced with permission from the American Chemical Society (ACS). Originally published in Journal of Medicinal Chemistry [9]. Copyright © 2024, American Chemical Society.

**Figure 5 ijms-26-11008-f005:**
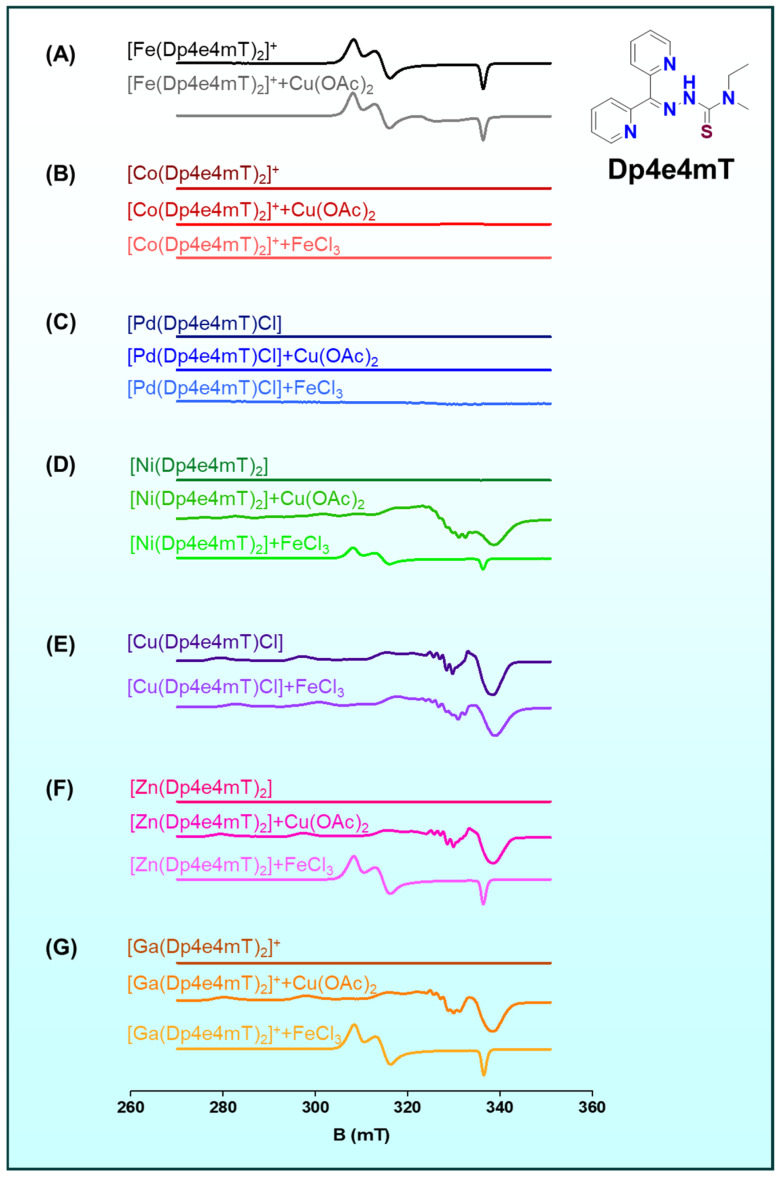
Complexes of Fe(III), Co(III), Pd(II), Ni(II), Cu(II), Zn(II), and Ga(III) with the ligand Dp4e4mT (shown at right) exhibitdistinct EPR profiles upon exposure to competing metals (Cu(II) or Fe(III)), defining the thermodynamic hierarchy of metal exchange. The emergence of Cu(II) signals confirms that Cu(II) efficiently displaces other metals, highlighting the high transmetalation propensity and redox flexibility of this system. (**A**) [Fe(Dp4e4mT)_2_]⁺ before and after addition of Cu(OAc)_2_; (**B**) [Co(Dp4e4mT)_2_]⁺ before and after addition of Cu(OAc)_2_ or FeCl_3_; (**C**) [Pd(Dp4e4mT)Cl] before and after addition of Cu(OAc)_2_ or FeCl_3_; (**D**) [Ni(Dp4e4mT)_2_] before and after addition of Cu(OAc)_2_ or FeCl_3_; (**E**) [Cu(Dp4e4mT)Cl] before and after addition of FeCl_3_; (**F**) [Zn(Dp4e4mT)_2_] before and after addition of Cu(OAc)_2_ or FeCl_3_; and (**G**) [Ga(Dp4e4mT)_3_]⁺ before and after addition of Cu(OAc)_2_ or FeCl_3_. Reproduced with permission from the Royal Society of Chemistry. Originally published in Chemical Science [5]. Copyright © 2024 Royal Society of Chemistry.

**Figure 6 ijms-26-11008-f006:**
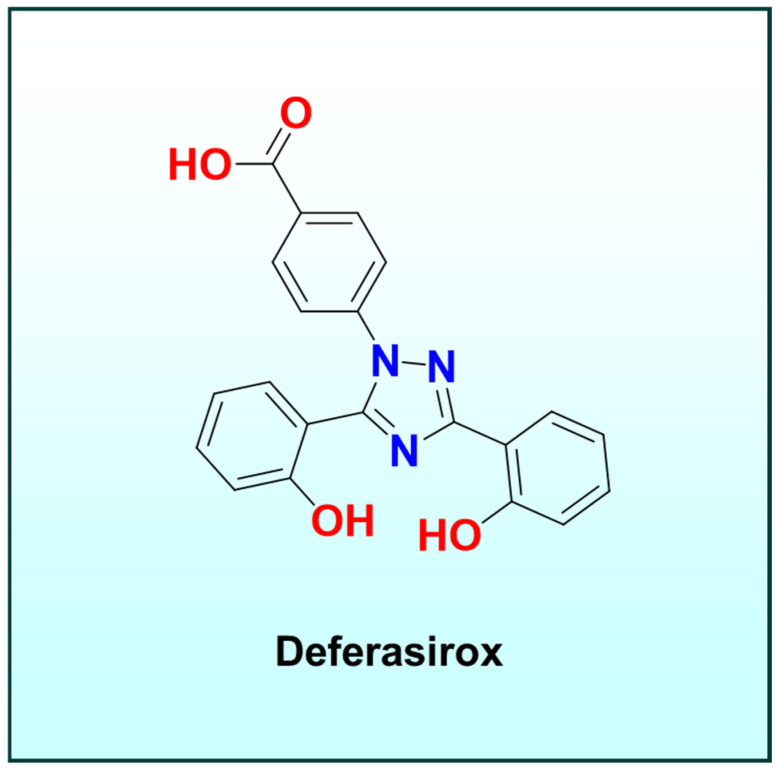
Chemical structure of deferasirox, an FDA-approved tridentate iron chelator composed of a bis-hydroxyphenyl–triazole scaffold.

**Figure 7 ijms-26-11008-f007:**
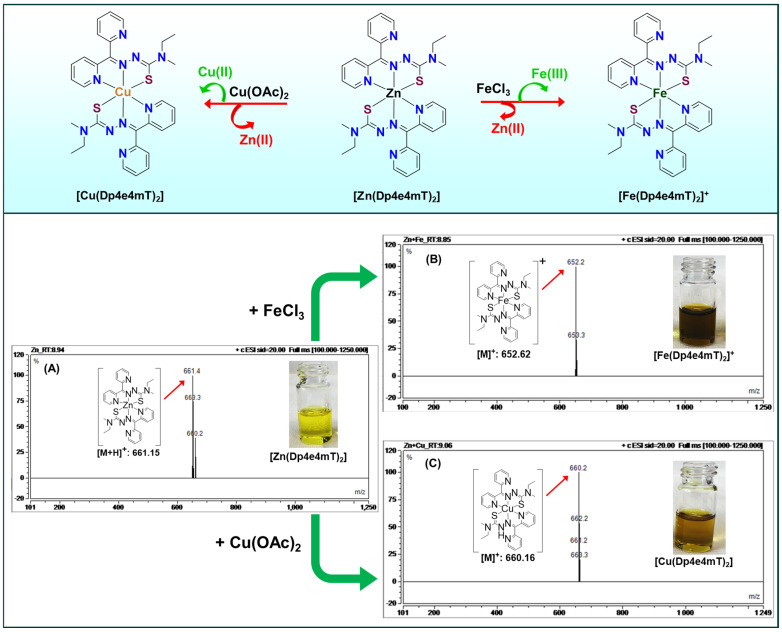
The Zn(II) complex readily undergoes metal exchange with Cu(II) to form [Cu(Dp4e4mT)_2_], and with Fe(III) to form [Fe(Dp4e4mT)_2_]^+^, as confirmed by ESI–MS. The parent [Zn(Dp4e4mT)_2_] peak shifts to the corresponding Cu(II)- and Fe(III)-containing species upon addition of Cu(OAc)_2_ or FeCl_3_, respectively. (**A**) ESI–MS spectrum of [Zn(Dp4e4mT)_2_] showing the parent ion at *m*/*z* 661.15. (**B**,**C**) ESI–MS spectra of transmetalated products obtained after addition of FeCl_3_ or Cu(OAc)_2_, showing formation of [Fe(Dp4e4mT)_2_]⁺ (*m*/*z* 652.62) and [Cu(Dp4e4mT)_2_] (*m*/*z* 660.16), respectively. Reproduced with permission from the Royal Society of Chemistry. Originally published in Chemical Science [5]. Copyright © 2024 Royal Society of Chemistry.

**Figure 8 ijms-26-11008-f008:**
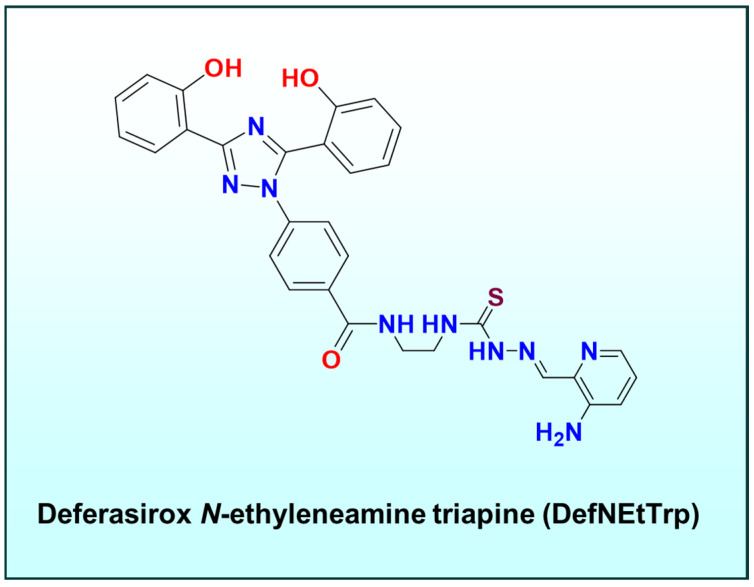
Chemical structure of the dual-chelator DefNEtTrp [18].

**Figure 9 ijms-26-11008-f009:**
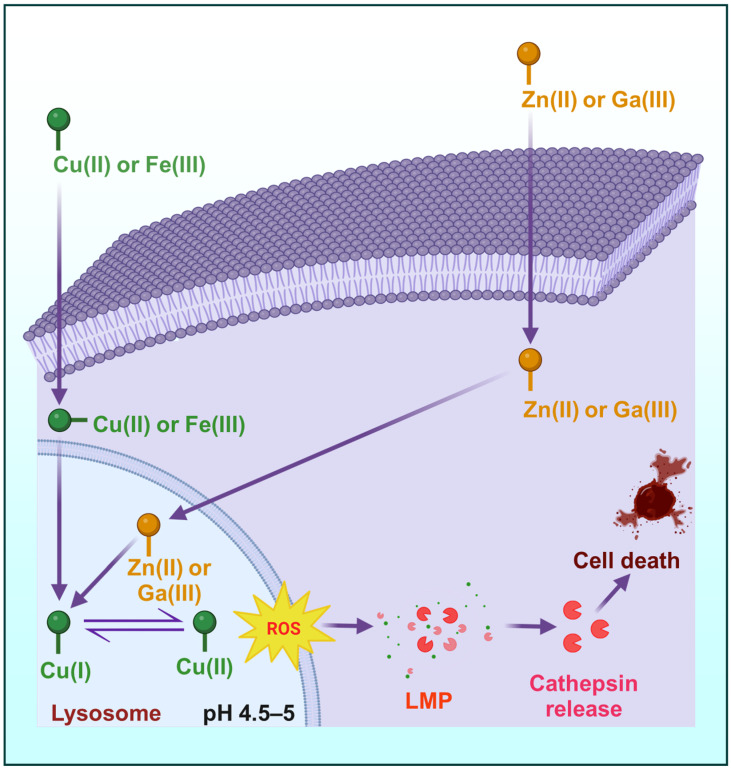
Schematic illustration of lysosomal transmetalation and redox-mediated cell death. Zn(II)- or Ga(III)-based complexes enter cells and undergo metal exchange with endogenous Cu(II) or Fe(III) within lysosomes (pH 4.5–5). The resulting Cu(II)–ligand species undergo redox cycling with Cu(I), generating ROS that destabilize LMP, leading to cathepsin release and apoptotic cell death.

**Figure 10 ijms-26-11008-f010:**
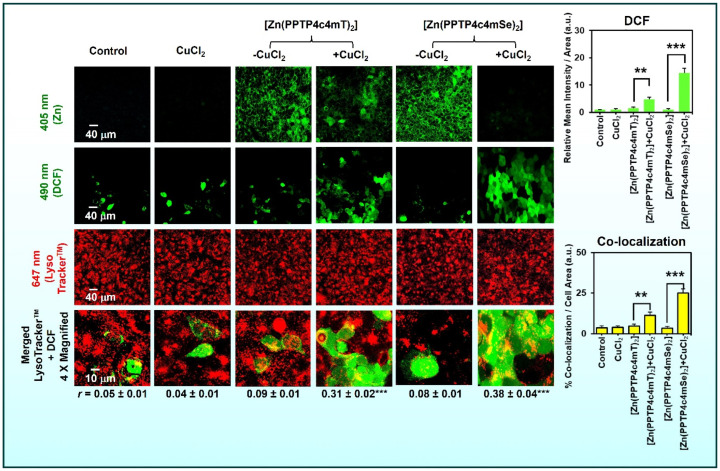
Confocal fluorescence microscopy illustrating the enhanced lysosomal transmetalation and ROS generation by the Zn(II)–selenosemicarbazone complex compared with its thiosemicarbazone analogue. Cells treated with [Zn(PPTP4c4mSe)_2_] displayed stronger DCF fluorescence and greater co-localization with LysoTracker™ Red than those treated with [Zn(PPTP4c4mT)_2_], indicating more efficient Cu(II) exchange and lysosomal ROS production. These findings support that selenium substitution facilitates Zn–Cu transmetalation and augments oxidative stress within lysosomes, correlating with the higher anticancer efficacy of the selenosemicarbazone system. Scale bars: 40 µm (main panels) and 10 µm (magnified images). Results are means ± SD (3). ** *p* < 0.01; *** *p* < 0.001 vs. the control or as indicated. Reproduced with permission from the American Chemical Society (ACS), originally published in Journal of the American Chemical Society [9]. Copyright © 2024 American Chemical Society.

**Figure 11 ijms-26-11008-f011:**
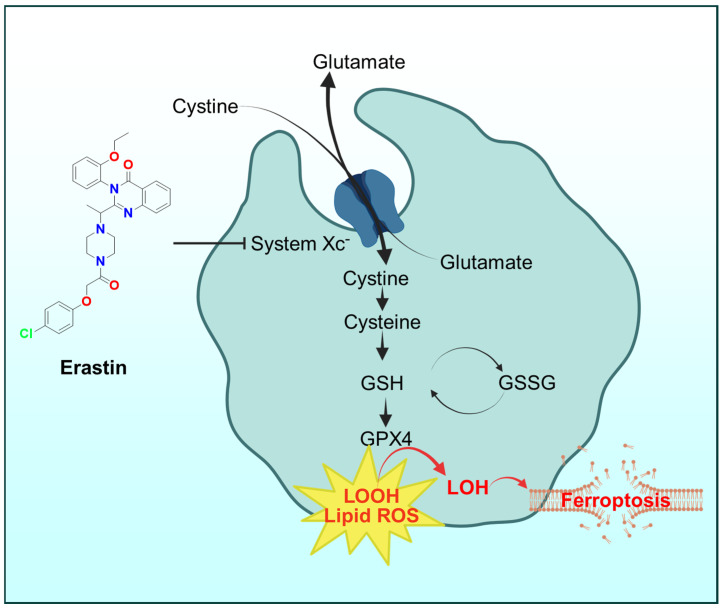
Mechanism of ferroptosis induction by erastin. Erastin inhibits the cystine/glutamate antiporter (System Xc^−^), blocking cystine import and depleting intracellular cysteine and GSH. The resulting loss of GSH disables GPX4, leading to accumulation of lipid hydroperoxides (LOOH) and lipid radicals (LO^•^), which trigger oxidative membrane damage and ferroptotic cell death.

**Table 1 ijms-26-11008-t001:** Representative redox potentials of biologically relevant metal couples (E° vs. NHE).

Metal Couple	Half-Reaction (Reduction Direction)	E° vs. NHE (V)	Biological Relevance
Fe^3+^/Fe^2+^	Fe^3+^ + e^−^ → Fe^2+^	+0.16 to +0.82	Redox-active; drives Fenton-like ROS formation [8].
Cu^2+^/Cu^+^	Cu^2+^ + e^−^ → Cu^+^	+0.15 to −0.25	Redox-active; supports ROS generation and redox cycling [8].
Zn^2+^/Zn	Zn^2+^ + 2e^−^ → Zn(s)	−1.2 to −1.4	Redox-inert; not involved in Fenton-type reactions [8,13].
Ga^3+^/Ga	Ga^3+^ + 3e^−^ → Ga(s)	−0.53	Redox-inert; mimics Fe(III) without ROS activity [5].
Ti^4+^/Ti^3+^	Ti^4+^ + e^−^ → Ti^3+^	−0.9	Weakly redox-active; unlikely to drive Fenton chemistry [83].

Note: Zn(s) and Ga(s) denote the metallic forms, as Zn^+^ and Ga^+^ are unstable in aqueous media.

## Data Availability

No new data were created or analyzed in this study. Data sharing is not applicable to this article.

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
