# Peer review of "Transmetalation in Cancer Pharmacology"

_ijms, 2025, doi:10.3390/ijms262211008_

Round 1

Reviewer 1 Report

Comments and Suggestions for Authors

This review integrates chemical and biological perspectives on transmetallation in cancer pharmacology, with a specific focus on metal-based ligand systems—particularly thioacylhydrazines—and their binding to Ti(IV), Fe(III), Co(III), Ni(II), Cu(II), Zn(II), and Pd(II). The review holds significant value and is recommended for publication after revisions. Minor issues requiring attention are as follows:

  1. The authors should clarify three key points: first, the administration routesof metal-based anticancer complexes; second, the water solubilityof these molecules; and third, whether the intrinsic toxicity of such small molecules will become a barrier limiting their clinical applications.
  2. Could the authors present the metal exchange processes of the metal-based anticancer complexes mentioned in this review in the form of chemical reaction equations? This would make the content of the review much clearer for readers to understand.
  3. Can this type of metal-based complex be applied in antibacterial applications? The authors are advised to supplement relevant discussions or explanations.
  4. Glutathione is abbreviated as GSH by the authors at the end of the manuscript; however, the full term "glutathione" is still used throughout the rest of the text. In this case, the purpose of abbreviating it later loses its significance. The authors should standardize the use of the abbreviation (e.g., define it at the first occurrence and use the abbreviation consistently thereafter).
  5. The formatting of abbreviations such as reactive oxygen species (ROS) is inconsistent, with the full term and abbreviation repeatedly appearing together throughout the manuscript. This leads to unnecessary redundancy, and the authors should unify the abbreviation usage in line with academic writing conventions.

Author Response

Referee 1: Recommendation: Minor Revision

 Q1. The authors should clarify three key points: first, the administration routes of metal-based anticancer complexes; second, the water solubility of these molecules; and third, whether the intrinsic toxicity of such small molecules will become a barrier limiting their clinical applications.

Revision: We thank the reviewer for highlighting these important translational considerations. The revised manuscript now includes an expanded discussion addressing (i) administration routes, (ii) aqueous solubility, and (iii) intrinsic toxicity of metal-based anticancer complexes.

(i) Administration routes:

Most ligand and metal-based agents (e.g., Dp44mT, Triapine®, and gallium maltolate) have been administered intravenously in preclinical and clinical studies to ensure full bioavailability and to prevent degradation in the gastrointestinal tract. However, the improved analogue DpC demonstrates high stability and oral bioavailability, which has enabled its progression to clinical trials. Furthermore, lipophilic and nanoformulated derivatives, including Zn(II)–thiosemicarbazone “stealth” complexes, show potential for oral or targeted delivery. These aspects are now clearly discussed in the revised manuscript.

(ii) Water solubility:

Many thiosemicarbazone analogues and their Cu(II)/Fe(III) complexes are poorly water-soluble, often requiring formulation using cyclodextrins, liposomes, or PEGylated carriers. The manuscript highlights how ligand modifications in the DpC and PPP44mT series enhance solubility and pharmacokinetics by improving polarity while maintaining optimal lipophilicity.

(iii) Intrinsic toxicity:

The expanded discussion also emphasizes that Cu(II) and Fe(III) complexes can induce off-target oxidative effects (e.g., hemoglobin or myoglobin oxidation), whereas redox-inert Zn(II) and Ga(III) complexes are generally well tolerated. The revised text discusses formulation and structural strategies including redox-inert prodrug design, controlled transmetalation, and liposomal encapsulation that mitigate toxicity while preserving anticancer potency.

Corresponding revisions have been made in the following sections of the manuscript:

Page 14:

 In addressing clinical translation, formulation and delivery strongly influence toxicity outcomes (111). Redox-active Cu(II) and Fe(III) complexes are generally administered intravenously in animal and human studies to avoid gastrointestinal degradation and ensure bioavailability (112). Their intrinsic redox activity can lead to hemoglobin oxidation and oxidative organ stress; however, careful dose optimization and the use of redox-inert or “stealth” analogues such as Zn(II)- and Ga(III)-thiosemicarbazones mitigate these effects (8). Moreover, formulation approaches including encapsulation in liposomes or PEGylated nanocarriers have been shown to reduce systemic toxicity while preserving tumor selectivity (113).

Page 22:

The clinical translation of metal-based complexes is closely tied to their solubility, formulation, and delivery route. Compounds such as Dp44mT have primarily been evaluated through intravenous administration (217), whereas its improved analogue DpC demonstrates high stability and oral bioavailability (36), enabling successful progression to clinical trials. Lipophilic analogues and nano formulations now further support oral or targeted delivery options. The water solubility of thiosemicarbazone ligands varies widely; cyclodextrin inclusion, liposomal dispersion, or introduction of polar substituents (e.g., morpholine, piperazine) have been effective strategies to enhance aqueous compatibility (218, 219). Addressing intrinsic toxicity remains essential, as Cu(II) and Fe(III) complexes may exhibit oxidative reactivity, whereas Ga(III) and Zn(II) complexes act as redox-inert prodrugs with improved tolerability. Collectively, these advances balance efficacy with safety to overcome barriers to clinical translation.

Q2. Could the authors present the metal exchange processes of the metal-based anticancer complexes mentioned in this review in the form of chemical reaction equations? This would make the content of the review much clearer for readers to understand.

Revision: We thank the reviewer for this helpful suggestion. To improve clarity and mechanistic understanding, the revised manuscript now explicitly presents the metal-exchange (transmetalation) reactions in the form of balanced chemical equations. These equations illustrate how Zn(II)–thiosemicarbazone complexes act as prodrugs that undergo metal exchange with biologically available Cu(II) or Fe(III) to generate redox-active cytotoxic species.

Accordingly, we have added representative reaction schemes in two locations:

Page 5 (Section 2.2, Transmetalation as a Design Principle):

……‘‘Representative transmetalation reactions:

(1) [Zn(DpC)2] + Cu2+ [Cu(DpC)2] + Zn2+

(2) [Zn(DpC)2] + Cu2+ [Cu(DpC)]+ + Zn2+ + DpC¯

These equations 1 and 2 illustrate the neutral and mono-ligand pathways, respectively, by which Zn(II)–thiosemicarbazone complexes convert into redox-active Cu(II) species under physiological conditions (9).’’

Page 9 (Section 2.3.4, Transmetalation of Zn(II) to Fe(III)):

‘‘(3) [Zn(Dp4e4mT)₂] + Fe³⁺ [Fe(Dp4e4mT)₂]⁺ + Zn²⁺’’

Q3. Can this type of metal-based complex be applied in antibacterial applications? The authors are advised to supplement relevant discussions or explanations.

Revision: We thank the reviewer for this valuable suggestion. Several metal-based thiosemicarbazone (TSC) complexes have demonstrated broad-spectrum antibacterial activity in addition to their anticancer effects. The mechanisms are analogous to those responsible for tumor cell death, involving metal chelation, disruption of microbial redox homeostasis, and reactive oxygen species (ROS) generation.

The revised manuscript now includes a concise discussion in Section 7 (Translational Considerations and Broader Applications, page 22) summarizing these antibacterial properties. It highlights that Cu(II), Fe(III), and Zn(II) TSC complexes disrupt bacterial metal metabolism and promote oxidative stress leading to membrane and DNA damage. Structural modifications that increase lipophilicity or facilitate nanocarrier incorporation enhance bacterial uptake and biofilm penetration. These findings, supported by recent reports (Refs 184, 186), indicate that the same redox-active principles exploited in oncology can also be harnessed to combat bacterial infections, including antibiotic-resistant strains.

The following statements have been added to the revised manuscript on page 22 (second paragraph):

‘‘Beyond oncology, several metal–thiosemicarbazone complexes also demonstrate antibacterial potential through mechanisms analogous to their anticancer activity. Cu(II) and Fe(III) complexes catalyze Fenton-type redox cycling, generating ROS that damage bacterial membranes, proteins, and DNA, while Zn(II) complexes can inhibit metalloenzymes critical for microbial survival. Incorporation of lipophilic substituents or formulation within nanocarriers enhances bacterial uptake and biofilm disruption. These properties, reported for several TSC analogues and their metal complexes, underscore the wider biomedical versatility of this chemical class and its promise for ad-dressing antibiotic-resistant infections (184, 186).’’

Q4. Glutathione is abbreviated as GSH by the authors at the end of the manuscript; however, the full term "glutathione" is still used throughout the rest of the text. In this case, the purpose of abbreviating it later loses its significance. The authors should standardize the use of the abbreviation (e.g., define it at the first occurrence and use the abbreviation consistently thereafter).

Revision: We thank the reviewer for this helpful observation. The abbreviation has now been standardized throughout the manuscript. Specifically, “glutathione (GSH)” is defined at its first occurrence in Section 2.3.1, page 8 (Transmetalation of Zn(II) to Cu(II)), and the abbreviation “GSH” is used consistently thereafter throughout the text.

Q5. The formatting of abbreviations such as reactive oxygen species (ROS) is inconsistent, with the full term and abbreviation repeatedly appearing together throughout the manuscript. This leads to unnecessary redundancy, and the authors should unify the abbreviation usage in line with academic writing conventions.

Revision: Thank you. We agree with the reviewer’s comment. The abbreviation “reactive oxygen species (ROS)” has now been standardized throughout the manuscript. It is defined at its first occurrence in Section 1, Pages 1-2 (Introduction), and thereafter, only the abbreviation “ROS” is used consistently in the main text and figure legends.

Reviewer 2 Report

Comments and Suggestions for Authors

In this review by Dharmasivam and Kaya, the authors outline the current innovations being pursued in using transmetallation as a chemical tool for cancer biology. The authors have a fairly exhaustive set of referencing and do a good job of surveying the current state of metal chelation and transmetallation and its biological activity in cancer research. However, the figures are challenging to read given the styling the authors selected (gold bonds) and contain many inconsistencies of formatting. Additionally, a lot of the data presented in the review could be more appropriately represented in a table, or graph. Finally, the authors need to specify the role of specific cancer cell lines rather than making broad statements about “cancers” as the activity of different metals in cancer cells can vary from cell lineage to cell lineage. 

Figure 1: While artistic, the golden structures are distracting and difficult to read. The authors should accompany this or replace this with a conventional ChemDraw file that more clearly depicts the structures of these compounds.

Section 3.1: Enterobactin is a natural product from the bacterial world and its siderophore activity has been repurposed for cancer. The authors should comment on this specifically and also discuss implications of other naturally-occurring siderophores. 

Figure 5: The structure of Dp4e4mT should be included in this figure, as it is not obvious to some readers who are not familiar with the author’s acronyms.

Section 3.3.3: A figure depicting deferasirox should be included in this section. Additionally, the authors should mention relative ligand-metal binding energies of Ti and Fe in this and in other systems mentioned. 

Section 4.2: In this section, the authors detail hypothetical reasons for why organometallic complexes selectively target tumor cells and tissues over healthy cells and tissues. Several of these claims are overly-generalized and it is unlikely that such claims can be drawn across all tumors. For example, the authors write, “Cancer cells typically have an elevated labile iron pool and higher free Cu levels due to hyperactive metabolism and upregulated metal import proteins”. The authors should cite the specific cancer cell lines from which such conclusions are drawn and should avoid generalizing such statements across all cancers, as specifics of metabolism of each cancer type remain variable in nature. 

Section 4, P. 12: The authors mention Fenton-like electron transfer chemistry as the potentially operative mechanism through which Cu(II) and Fe(III) exhibit their cytotoxic effects. The authors should include a table of redox potentials of common metals covered in this review and should draw generalized conclusions on which metals based on redox potential are most likely to be involved in Fenton-like processes and reactions with biological material. 

Figure 10: The chemical structure of Erastin is of a different format than the rest of the chemical structures listed in the paper, with color coded atoms and different bond formats. I would recommend the authors uniformize chemical structure formatting.

A generally well known episode in medicinal chemistry where Cu has played a pivotal role is in the discovery that copper (II) chelation on the putative small molecule TET inhibitor Bobcat339 (Chua, et al. ACS Med Chem. Lett. 2019; Weirath, et al. ACS Med Chem. Lett. 2022) should perhaps be included in this review as well as it presents a mechanistic role of copper that is distinct from what is already presented in the paper. 

Author Response

Referee 2: Recommendation: Minor Revision

Q1. Figure 1: While artistic, the golden structures are distracting and difficult to read. The authors should accompany this or replace this with a conventional ChemDraw file that more clearly depicts the structures of these compounds.

Revision: We thank the reviewer for this constructive feedback. The original artistic version of Figure 1 has been replaced with a conventional ChemDraw-rendered figure that clearly depicts the molecular structures of the discussed compounds. The new version provides accurate bond connectivity, labeling, and clarity while maintaining visual consistency with the remaining figures in the manuscript.

Q2. Section 3.1: Enterobactin is a natural product from the bacterial world and its siderophore activity has been repurposed for cancer. The authors should comment on this specifically and also discuss implications of other naturally-occurring siderophores.

Revision: We thank the reviewer for this insightful suggestion. The revised manuscript now includes an expanded discussion in Section 2.1 (Beyond TSCs and Other Metal–Ligand Systems, page 4) addressing the role of enterobactin and other naturally occurring siderophores in cancer-related metal chelation. Specifically, the new paragraph highlights that enterobactin, a bacterial catecholate siderophore, has been repurposed in oncology due to its exceptionally high Fe(III)-binding affinity and capacity to disrupt tumor iron metabolism. Enterobactin and its synthetic analogues were shown to suppress cancer cell proliferation through intracellular iron depletion and redox modulation, demonstrating how microbial iron-acquisition chemistry can be leveraged for therapeutic design.

The paragraph also briefly discusses other natural siderophores, including deferoxamine and pyoverdine, which similarly interfere with tumor iron homeostasis and serve as bioinspired scaffolds for developing new anticancer chelators.

The following statement has been added to the revised manuscript on page 4:

‘‘Notably, enterobactin, a bacterial catecholate siderophore, has been repurposed in oncology: enterobactin and synthetic analogues deplete intracellular Fe(III) and suppress proliferation by iron starvation and redox modulation, illustrating how microbial iron-acquisition chemistry can be leveraged against tumors (45-47). Related naturally occurring siderophores, including desferrioxamine and pyoverdine (Figure 2C), similarly perturb tumor iron homeostasis and provide bioinspired templates for anticancer chelators (48-54).’’

Q3. Figure 5: The structure of Dp4e4mT should be included in this figure, as it is not obvious to some readers who are not familiar with the author’s acronyms.

Revision: Thank you. We agree, and the revised Figure 5 now includes the chemical structure of Dp4e4mT, ensuring that all compounds represented in the figure are clearly identified.

Q4. Section 3.3.3: A figure depicting deferasirox should be included in this section. Additionally, the authors should mention relative ligand-metal binding energies of Ti and Fe in this and in other systems mentioned.

Revision: We thank the reviewer for this helpful suggestion. The revised manuscript now includes the chemical structure of deferasirox as Figure 6 in Section 2.3.3, improving clarity and visual representation of this clinically relevant tridentate chelator.

We have also expanded the discussion to qualitatively describe ligand–metal interactions between Ti(IV) and Fe(III). Consistent with published data and hard–soft acid–base (HSAB) principles, deferasirox exhibits stronger coordination with Fe(III) through its oxygen donor atoms, while Ti(IV) complexes are relatively more labile under physiological conditions. This difference rationalizes the observed Fe(III) to Ti(IV) transmetalation behavior in biological systems.

The following statement has been added to the revised manuscript on page 9:

‘‘Consistent with hard–soft acid–base (HSAB) principles, deferasirox (Figure 6) is an O-donor tridentate ligand that binds Fe(III) more strongly than Ti(IV), making Fe(III) substitution thermodynamically favored once inside the cell (83, 84). This preference underpins the observed Fe(III) to Ti(IV) transmetalation behavior in biological systems (83, 85).’’

Q5. Section 4.2: In this section, the authors detail hypothetical reasons for why organometallic complexes selectively target tumor cells and tissues over healthy cells and tissues. Several of these claims are overly-generalized and it is unlikely that such claims can be drawn across all tumors. For example, the authors write, “Cancer cells typically have an elevated labile iron pool and higher free Cu levels due to hyperactive metabolism and upregulated metal import proteins”. The authors should cite the specific cancer cell lines from which such conclusions are drawn and should avoid generalizing such statements across all cancers, as specifics of metabolism of each cancer type remain variable in nature.

Revision: We thank the reviewer for this valuable comment and agree that metal homeostasis differs among cancer types. The revised manuscript now clarifies that these observations apply to representative cancer models rather than all tumors. Specific examples and supporting references have been added to demonstrate the variability and context-dependence of metal dysregulation in cancer.

The following statement has been added to the revised manuscript on page 12:

‘‘3.2.1. Elevated intracellular metals in tumors.

Several cancer models exhibit elevated labile iron and increased free copper levels associated with hyperactive metabolism and upregulated metal-import pathways. For example, breast (MCF-7, MDA-MB-231) (92, 93), prostate (PC-3, DU145) (94), and neuroepithelioma (SK-N-MC) (94) cells show higher TfR1 and CTR1 expression and greater labile Fe and Cu levels compared with non-malignant counterparts (95-98). In contrast, normal cells tightly regulate and minimize free metal ions (99). Therefore, in such contexts, a transmetalation-dependent drug encounters more endogenous “fuel” for activation, enhancing selectivity (100).’’

Q6. Section 4, P. 12: The authors mention Fenton-like electron transfer chemistry as the potentially operative mechanism through which Cu(II) and Fe(III) exhibit their cytotoxic effects. The authors should include a table of redox potentials of common metals covered in this review and should draw generalized conclusions on which metals based on redox potential are most likely to be involved in Fenton-like processes and reactions with biological material.

Revision: We thank the reviewer for this valuable suggestion. The revised manuscript now includes Table 1 (page 11) summarizing representative standard reduction potentials (E° vs. NHE) for the key metal couples discussed. As noted in the table, biological factors such as pH, ligand coordination, and ionic strength can shift these values, so the data are intended for orientation and comparison only. Consistent with these potentials and our mechanistic discussion, Fe(III)/Fe(II) and Cu(II)/Cu(I) lie within the biologically accessible redox range and are most likely to mediate Fenton-like ROS generation, whereas Zn(II) and Ga(III) are redox-inert, and Ti(IV) shows limited redox activity under physiological conditions.

The following statement and table have been added to the revised manuscript (page 11):

‘‘To clarify redox trends, representative redox potentials are summarized in Table 1; only Fe3+/Fe2+ and Cu2+/Cu+ lie within the biological redox window, enabling Fenton-like ROS formation, while Zn2+/Zn and Ga3+/Ga remain redox-stable and Ti4+/Ti3+ is only weakly active.’’

Q7. Figure 10: The chemical structure of Erastin is of a different format than the rest of the chemical structures listed in the paper, with color coded atoms and different bond formats. I would recommend the authors uniformize chemical structure formatting.

Revision: We thank the reviewer for this helpful suggestion. To ensure consistency, the other chemical structures throughout the manuscript have been re-drawn and reformatted to match the Erastin structure style presented in Figure 10 (now Figure 11). All figures now use a uniform ChemDraw format with consistent bond thickness, alignment, and color scheme, achieving a standardized visual presentation across the manuscript.

Q8. A generally well known episode in medicinal chemistry where Cu has played a pivotal role is in the discovery that copper (II) chelation on the putative small molecule TET inhibitor Bobcat339 (Chua, et al. ACS Med Chem. Lett. 2019; Weirath, et al. ACS Med Chem. Lett. 2022) should perhaps be included in this review as well as it presents a mechanistic role of copper that is distinct from what is already presented in the paper.

Revision: We thank the reviewer for this thoughtful suggestion. The example of Bobcat339 indeed illustrates an interesting Cu(II)-dependent enzyme inhibition mechanism. However, as the present review focuses specifically on transmetalation and redox-based mechanisms of metal complexes in cancer therapy, the inclusion of Bobcat339, which operates through Cu(II)-mediated TET enzyme modulation, would fall outside the defined chemical and mechanistic scope of this article. We therefore respectfully chose not to include this case, while fully acknowledging its relevance within the broader field of copper biochemistry.
